# PRISM: Efficient Test-Time Scaling via Hierarchical Search and Self-Verification for Discrete Diffusion Language Models

**Jinbin Bai** [1 2]  **Yixuan Li** [3]  **Yuchen Zhu** [4]  **Yi Xin** [5]  **Qingyu Shi** [6]  **Aosong Feng** [7]  **Xiaohong Liu** [5]  **Molei Tao** [4]
**Jianru Xue** [3]  **Xiangtai Li** [6]  **Ming-Hsuan Yang** [8]

## Abstract

Inference-time compute has re-emerged as a practical way to improve LLM reasoning. Most test-time scaling (TTS) algorithms rely on autoregressive decoding, which is ill-suited to discrete diffusion language models (dLLMs) due to their parallel decoding over the entire sequence. As a result, developing effective and efficient TTS methods to unlock dLLMs' full generative potential remains an underexplored challenge. To address this, we propose **PRISM** (**P**runing, **R**emasking, and **I**ntegrated **S**elf-verification **M**ethod), an efficient TTS framework for dLLMs that (i) performs Hierarchical Trajectory Search (HTS) which dynamically prunes and reallocates compute in an early-to-mid denoising window, (ii) introduces Local branching with partial remasking to explore diverse implementations while preserving high-confidence tokens, and (iii) replaces external verifiers with Self-Verified Feedback (SVF) obtained via self-evaluation prompts on intermediate completions. Across four math reasoning and code generation benchmarks on three dLLMs, including LLaDA 8B Instruct, Dream 7B Instruct, and LLaDA 2.0-mini, our PRISM achieves a favorable performance-efficiency trade-off, matching best-of-$N$ performance with substantially fewer function evaluations (NFE). The code is released at https://github.com/viiika/Prism.

## 1. Introduction

The scaling laws of Large Language Models (LLMs) (Achiam et al., 2023) have traditionally fo-cused on training-time compute by increasing model parameters and dataset size. Recently, test-time scaling (TTS), which allocates additional compute at inference time to perform exploration, verification, and selection, has become a dominant paradigm for improving complex reasoning without retraining (Jaech et al., 2024a). However, most prior TTS work (Muennighoff et al., 2025; Wei et al., 2022; Wang et al., 2022; Brown et al., 2024; Jain et al., 2024; Snell et al., 2024), is built around autoregressive (AR) decoding, where search expands a left-to-right tree and early mistakes are difficult to correct without backtracking.

Discrete diffusion language models (dLLMs) such as LLaDA (Nie et al., 2025; Bie et al., 2025), Seed Diffusion (Song et al., 2025), Mercury (Khanna et al., 2025), and Gemini Diffusion (Google DeepMind, 2025) represent a fundamental departure from the autoregressive (AR) paradigm. By generating sequences with iterative denoising from a masked state, dLLMs utilize global bidirectional context at every generation step. This parallel, non-autoregressive generation process theoretically makes dLLMs superior candidates for planning and self-correction.

Previous test-time scaling methods typically allocate additional inference compute along two complementary axes (Muennighoff et al., 2025): *(i) length scaling* and *(ii) width scaling*. Length scaling extends the reasoning budget by generating longer responses (*e.g.*, chain-of-thought (Wei et al., 2022)) or increasing iterative refinement steps, whereas width scaling broadens the hypothesis space by exploring multiple candidate trajectories. Notably, increasing the number of denoising steps is often a less practical lever for dLLMs. In current dLLMs implementations, the default inference schedule is typically already saturated, with the number of generation steps commonly tied to the target sequence length, unlike image generation models where thousands of image tokens can often be predicted with only 10-50 inference steps (Chang et al., 2022; Li et al., 2025; Bai et al., 2024; Shi et al., 2025; Yang et al., 2025; Xin et al., 2025). Consequently, we focus on scaling width by generating $N$ diverse trajectories and selecting the best one to increase the likelihood of finding an optimal answer. Concurrent work, such as HEX (Lee et al., 2025a),

[1]National University of Singapore [2]Collov Labs [3]Xi'an Jiaotong University [4]Georgia Institute of Technology [5]Shanghai Innovation Institute [6]Peking University [7]Yale University [8]UC Merced. Correspondence to: Jinbin Bai <jinbin.bai@u.nus.edu>.

*Proceedings of the 43rd International Conference on Machine Learning*, Seoul, South Korea. PMLR 306, 2026. Copyright 2026 by the author(s).

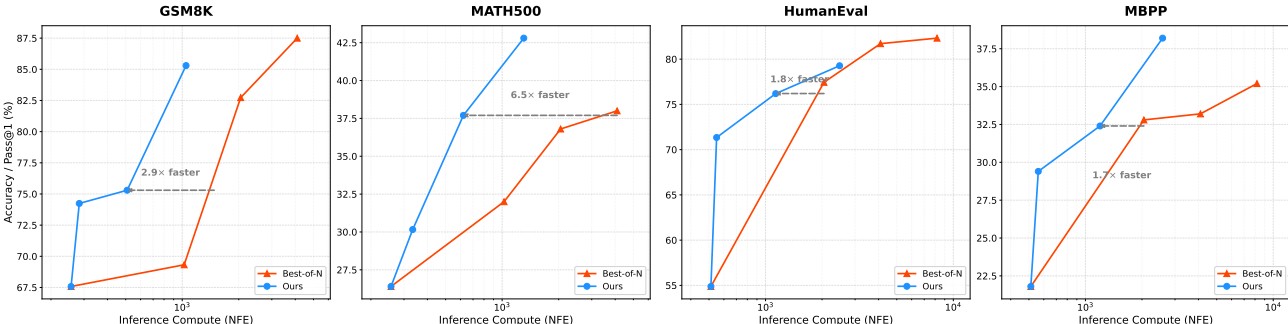

*Figure 1.* Comparison between Best-of-N and PRISM on LLaDA-8B-Instruct. The red curve illustrates Best-of-N scaling, while the blue curve depicts PRISM scaling, with a dashed line indicating the difference in inference compute (NFE) with comparable accuracy.

successfully explores a related width-scaling paradigm by ensembling across diverse semi-autoregressive block schedules. While this schedule-induced diversity yields strong reasoning gains, it fundamentally relies on exhaustive parallel decoding where all candidate trajectories must run to completion. Realizing this potential is non-trivial: naive best-of-$N$ search or HEX (Lee et al., 2025a) for dLLMs is computationally prohibitive, since evaluating $N$ trajectories over $T$ denoising steps requires $O(NT)$ function evaluations (NFE), and standard external verifiers further introduce substantial overhead (*e.g.*, GPU memory).

To address these bottlenecks, we introduce **PRISM**, an efficient test-time scaling framework tailored for dLLMs. First, we propose **Hierarchical Trajectory Search** (HTS), which employs a geometric decay schedule to progressively prune the active trajectory set and reallocate compute *within the early-to-mid denoising window* when the high-level logic skeleton is formed. Second, we introduce **local branching via partial re-masking**, an exploration operator that preserves high-confidence tokens as a stable "logic skeleton" while selectively re-masking low-confidence positions to explore diverse implementations under the same solution plan. Third, we replace external reward models with **Self-Verified Feedback**(SVF): we reuse the same dLLM as a lightweight binary verifier by applying a dedicated `Yes`/`No` self-evaluation prompt to intermediate completions, enabling verifier-guided pruning and selection with minimal additional overhead. This design yields a favorable compute profile: while best-of-$N$ incurs $O(NT)$ denoising cost, HTS rapidly contracts the trajectory pool from $N$ to $K < N$ after a short warm-up, resulting in near-linear scaling in NFE, approximately $O(N + KT)$ in practice.

Our contributions are summarized as follows:

- We propose PRISM, an efficient TTS framework for dLLMs that integrates Hierarchical Trajectory Search (HTS), local branching with partial re-masking, and Self-Verified Feedback (SVF) to enable adaptive exploration and selection without external reward models.

- Across four math and code benchmarks on three dLLMs, PRISM yields consistent gains over $N$=1 and matches or approaches Best-of-$N$ baselines under markedly reduced denoising compute (NFE), demonstrating a strong performance-efficiency trade-off.

## 2. Related Work

**Discrete Diffusion Language Models.** Discrete diffusion language models (dLLMs) (Khanna et al., 2025; Google DeepMind, 2025) replace left-to-right autoregressive decoding (Achiam et al., 2023; Hurst et al., 2024; Team et al., 2023) with a Markovian denoising process over token sequences. A canonical formulation builds on D3PM-style categorical diffusion(Austin et al., 2021a; Campbell et al., 2022; Lou et al., 2023), where a forward corruption chain is specified by time-dependent transition matrices and a learned reverse process iteratively denoises toward natural text. Two corruption families are most widely used. *Uniform* transitions (Schiff et al., 2024; Sahoo et al., 2025) mix tokens toward a uniform stationary distribution, offering a conceptually clean categorical analogue of Gaussian diffusion. *Absorbing-state* (*a.k.a.* masked) transitions (Ou et al., 2024; Sahoo et al., 2024; Shi et al., 2024) instead map tokens into a special absorbing symbol (typically [MASK]), yielding the masked diffusion model that aligns naturally with masked language modeling and admits particularly simple training and sampling rules.

Building on these foundations, subsequent work focused on simplifying objectives (*e.g.*, reweighted denoising cross-entropy (Chang et al., 2022)) and scaling architectures (Bai et al., 2024; Shi et al., 2025; Bai et al., 2025) to modern LLM regimes. Recent dLLMs (Nie et al., 2025; Bie et al., 2025; Khanna et al., 2025) demonstrate competitive performance on code and reasoning benchmarks while enabling non-autoregressive refinement and global bidirectional conditioning at each denoising step. Several works study how to obtain strong dLLMs at scale, either by training from scratch (Nie et al., 2025), or adopting a block-diffusion in-

terface (Arriola et al., 2025), or by converting pretrained autoregressive backbones into diffusion LMs (Gong et al., 2024; Ye et al., 2025). While recent works have attempted to scale discrete diffusion models (Huang et al., 2025a; Chen et al., 2025; Wang et al., 2025; Lee et al., 2025c), they often achieve only marginal performance gains or require significant computational overhead.

Our work operates in both inference settings and resolves a complementary question: how to allocate test-time compute effectively under multi-step denoising dynamics, without relying on external verifiers.

**Test-Time Scaling and Verification.** Test-time scaling (Wei et al., 2022; Wang et al., 2022; Brown et al., 2024; Jaech et al., 2024b; Muennighoff et al., 2025; Snell et al., 2024) studies how to convert additional inference-time computation into higher task accuracy by generating, refining, and selecting among multiple trajectories. Existing methods can be organized by their compute allocation pattern: *parallel* scaling expands a set of independent candidates and selects or aggregates them (*e.g.*, best-of-$N$, self-consistency with majority voting) (Irvine et al., 2023; Brown et al., 2024; Snell et al., 2024; Wang et al., 2022); *sequential* scaling iteratively revises a small number of evolving solutions (*e.g.*, self-refinement and correction loops) (Gou et al., 2023; Yao et al., 2022; Muennighoff et al., 2025); and *search-based* scaling adaptively expands and prunes a trajectory set under a scoring rule (*e.g.*, tree-style or MCTS-style deliberation) (Yao et al., 2023; Huang et al., 2025a). In all cases, the key algorithmic question is how to allocate compute adaptively by deciding which trajectories to keep exploring or to stop.

Verification provides the control signal that enables selection and pruning decisions. Prior work commonly distinguishes *outcome verification* (ORMs), which evaluates final answers using learned judges/reward models (Cobbe et al., 2021), self-consistency/voting (Wang et al., 2022), tool-assisted checks (Gou et al., 2023), or task-specific executors (especially effective in code) (Lee et al., 2025b), from *process verification* (PRMs) (Lightman et al., 2023; Yao et al., 2023), which scores intermediate states or step-wise progress to guide branching and pruning during search.

While PRMs have enabled effective tree-search for autoregressive decoding, they are typically trained on well-formed textual prefixes. For discrete diffusion language models (dLLMs) (Nie et al., 2025; Bie et al., 2025), intermediate denoising states are partially masked and do not follow a left-to-right prefix structure, which can make direct application of standard PRMs brittle or ill-calibrated. Moreover, in dLLMs each "candidate" often corresponds to a full denoising trajectory, so naive trajectory scaling can be computationally inefficient. These considerations motivate diffusion-

aligned TTS in which (i) the scoring signal remains meaningful on partially denoised states and (ii) computation is concentrated on the structure-formation stage rather than uniformly spread across steps. Our method follows this direction by using a lightweight self-verification score derived from the model's Yes/No confidence under a dedicated verification prompt and coupling it with hierarchical trajectory search for budgeted allocation.

**Relation to PG-DLM and SMC-style inference.** Dang et al. (2025) formulate inference-time scaling for diffusion language models as reward-tilted probabilistic inference and introduce PG-DLM, which constructs a Markov chain over full denoising trajectories using a conditional sequential Monte Carlo (SMC) kernel. PRISM shares with SMC-style methods a high-level population-based structure: maintaining multiple trajectories, scoring them, selecting promising candidates, and diversifying the remaining search pool. However, the underlying objectives and operators are different. PG-DLM targets a reward-weighted trajectory distribution and uses importance-weighted resampling within a probabilistic inference framework, thereby enabling convergence analysis under its assumptions. By contrast, PRISM is a budgeted combinatorial search procedure for verifiable reasoning tasks. SVF scores are used as heuristic ranking signals rather than density-ratio importance weights; HTS performs sparse top-$S$ pruning instead of resampling at every denoising step; and partial remasking locally mutates low-confidence positions rather than duplicating weighted particles. Thus, PG-DLM emphasizes sampling from a reward-tilted distribution while preserving generation quality, whereas PRISM emphasizes NFE-efficient allocation of search compute under a fixed inference budget.

This distinction is also relevant to particle-based inference-time scaling for autoregressive LLMs (Puri et al., 2025). In autoregressive decoding, each step conditions on a fixed prefix, and maintaining broad sampling diversity can be critical for mode coverage. In dLLMs, early denoising states are highly entropic and partially specified, so uniformly completing all candidates can waste compute on weak trajectories. Our entropy analysis and empirical results suggest that, for dLLMs, concentrating compute on promising trajectories during the early-to-mid denoising window is an effective alternative to exhaustive width scaling.

## 3. Method

### 3.1. Preliminaries: Discrete Diffusion Language Models

**Notation.** Let $\mathbf{z}_0 = (z_{0,1}, \ldots, z_{0,L}) \in [K]^L$ denote a length-$L$ token sequence over a vocabulary of size $K$. Let $\mathbf{e}(k) \in \{0, 1\}^K$ be the one-hot vector for token $k$, and let $\mathbf{1} \in \mathbb{R}^K$ denote the all-ones vector. We use the symbol $m$ (*e.g.*, [MASK]) to denote the special absorbing mask state

and write $\mathbf{e}_m \triangleq \mathbf{e}(m)$ for its one-hot vector. The diffusion timestep is $t \in \{1, \ldots, T\}$. When conditioning on a prompt, we denote it by $c$.

**Masked diffusion models.** Masked diffusion models (MDM) (also known as absorbing-state discrete diffusion models) are an especially effective variant of discrete diffusion models. MDM employs a forward process where the clean data sequences are progressively replaced with the mask token [MASK]. Formally speaking, the forward process follows the transition kernel

$$q(\mathbf{z}_t \mid \mathbf{z}_0, c) = \prod_{i=1}^{L} q_{t|0}(z_{t,i}|z_{0,i}),$$

$$q_{t|0}(z_{t,i} \mid z_{0,i}) = \mathrm{Cat}\big(z_{t,i}; \alpha_t\, \mathbf{e}(z_{0,i}) + (1-\alpha_t)\mathbf{e}_m\big),$$

where $\alpha_t$ is a monotonic mask-noising schedule. Recent works have shown that the training objectives can be directly related to optimizing an ELBO of the data log likelihood, given by

$$\mathcal{L}(\theta) = \mathbb{E}_{t,\mathbf{z}_0,\mathbf{z}_t} \left[ w(t) \sum_{i:z_{t,i}=m} \big( -\log \tilde{p}_\theta(z_{0,i} \mid \mathbf{z}_t, c, t) \big) \right],$$

**Inference through Block Diffusion.** Masked diffusion language model inference can be performed by iteratively unmasking tokens from a sequence of masks. Here, we adopt block diffusion decoding, an effective variant of such a sampling procedure, where a length-$L$ sequence is partitioned into $B = L/M$ contiguous blocks of size $M$ (*e.g.*, $L = 256$, $M = 32$). Generation proceeds block-by-block, in a left-to-right manner: once a block is finalized, it is treated as a fixed prefix, and the model moves to the next block.

Formally, at block index $b \in \{1, \ldots, B\}$, we maintain a partially specified state $\mathbf{z}_t^{(b)} \in (\mathcal{V} \cup \{m\})^L$ where blocks $1, \ldots, b-1$ are already committed tokens, while the current block $b$ is denoised from a fully masked initialization. Specifically, we start each block with

$$\mathbf{z}_T^{(b)} = \big[ \mathbf{x}^{(<b)},\ [m]^M,\ [m]^{L-bM} \big],$$

and iteratively update the tokens within the current block for $t = T, \ldots, 1$. Although the model predicts logits for all positions at every step (via a $\mathbf{z}_0$-prediction head), the sampling schedule commits only the current block, keeping the previously generated blocks fixed. After $T$ denoising steps, we finalize block $b$ and advance to $b+1$ until all blocks are generated.

**Sampling interface.** Even though $\mathbf{z}_t^{(b)}$ contains masked positions (within the current and future blocks) and is not a

---

**Algorithm 1** PRISM inference via Hierarchical Trajectory Search (HTS) and Self-Verified Feedback (SVF).

**Require:** Prompt $c$; dLLM denoiser $\mathcal{C}_\theta$; total steps $T$; initial width $N$; pruning window ratios $W = [w_{\min}, w_{\max}]$ (normalized by $T$); decay factor $d > 1$; pruning interval $i$; survivor width $S$; final target width $K$.
**Ensure:** Final completion $\hat{\mathbf{z}}_0$.
1: **Initialization.**
2: $T_p \leftarrow \lceil w_{\max} \cdot T \rceil$; $T_r \leftarrow \lceil w_{\min} \cdot T \rceil$.
3: Initialize $\mathcal{P}_T \leftarrow \{\mathbf{z}_T^{(n)}\}_{n=1}^N$ with $\mathbf{z}_T^{(n)} = [\mathrm{MASK}]^L$.
4: **Stage I: Stochastic exploration** ($T_p < t \leq T$).
5: **for** $t = T, T-1, \ldots, T_p + 1$ **do**
6: $\quad \mathcal{P}_{t-1} \leftarrow \{\mathrm{DenoiseStep}(\mathcal{C}_\theta, \mathbf{z}_t, c, t) \mid \mathbf{z}_t \in \mathcal{P}_t\}$.
7: **end for**
8: **Stage II: Progressive thinning** ($T_r < t \leq T_p$).
9: $r \leftarrow 0$. // prune and branch iff $r = 0$
10: **for** $t = T_p, T_p - 1, \ldots, T_r + 1$ **do**
11: $\quad$ **if** $r = 0$ **then**
12: $\quad\quad M_{t-1} \leftarrow \max\big(\lceil N \cdot d^{-(T_p - (t-1))} \rceil,\ K\big)$.
13: $\quad\quad$ // target width after pruning at step $t$
14: $\quad\quad \mathrm{score}(\mathbf{z}_t) \leftarrow \Phi_{\mathrm{SVF}}(\mathbf{z}_t; c)$ **for all** $\mathbf{z}_t \in \mathcal{P}_t$.
15: $\quad\quad \mathcal{S}_t \leftarrow \mathrm{TopS}(\mathcal{P}_t, S; \mathrm{score})$. // select top-$S$ seeds
16: $\quad\quad b_t \leftarrow \left\lceil \frac{M_{t-1}}{S} \right\rceil$. // children per survivor
17: $\quad\quad \mathcal{C}_t \leftarrow [\,]$.
18: $\quad\quad$ **for each** seed $\mathbf{z}_t \in \mathcal{S}_t$ **do**
19: $\quad\quad\quad$ **for** $j = 1$ **to** $b_t$ **do**
20: $\quad\quad\quad\quad \tilde{\mathbf{z}}_t \leftarrow \mathrm{LocalBranch}(\mathbf{z}_t, c, t)$.
21: $\quad\quad\quad\quad$ // local branching via partial remasking
22: $\quad\quad\quad\quad$ append $\mathrm{DenoiseStep}(\mathcal{C}_\theta, \tilde{\mathbf{z}}_t, c, t)$ to $\mathcal{C}_t$.
23: $\quad\quad\quad$ **end for**
24: $\quad\quad$ **end for**
25: $\quad\quad r \leftarrow i$. // wait $i$ steps before next pruning/branching
26: $\quad$ **else**
27: $\quad\quad \mathcal{P}_{t-1} \leftarrow \{\mathrm{DenoiseStep}(\mathcal{C}_\theta, \mathbf{z}_t, c, t) \mid \mathbf{z}_t \in \mathcal{P}_t\}$.
28: $\quad\quad r \leftarrow r - 1$.
29: $\quad$ **end if**
30: **end for**
31: $\mathcal{P}_{T_r} \leftarrow \mathrm{Truncate}(\mathcal{P}_{T_r}, K)$. // ensure final target width $K$ before refinement
32: **Stage III: Final refinement** ($1 \leq t \leq T_r$).
33: **for** $t = T_r, T_r - 1, \ldots, 1$ **do**
34: $\quad \mathcal{P}_{t-1} \leftarrow \{\mathrm{DenoiseStep}(\mathcal{C}_\theta, \mathbf{z}_t, c, t) \mid \mathbf{z}_t \in \mathcal{P}_t\}$.
35: $\quad$ **if** all $\mathbf{z} \in \mathcal{P}_{t-1}$ satisfy STOPCOND **then**
36: $\quad\quad$ **break** // e.g., no remaining MASK in the active window or an end-of-answer marker is detected
37: $\quad$ **end if**
38: **end for**
39: $\hat{\mathbf{z}}_0 \leftarrow \mathrm{SelectFinal}(\mathcal{P}_0)$. // majority voting
40: **return** $\hat{\mathbf{z}}_0$.

---

complete answer. The $\mathbf{z}_0$-prediction head provides a natural completion interface that yields a full hypothesis at any step:

$$\hat{\mathbf{z}}_0^{(i)} = \mathcal{C}_\theta(\mathbf{z}_t^{(b,i)}, c, t), \tag{1}$$

where $\mathcal{C}_\theta$ is instantiated by token-wise argmax $\hat{z}_{0,j} = \arg\max_k \tilde{p}_\theta(z_{0,j} = k \mid \mathbf{z}_t, c, t)$. We apply verification and trajectory selection on completed hypotheses $\hat{\mathbf{z}}_0^{(i)}$, while continuing denoising within the current block state $\mathbf{z}_t^{(b,i)}$ to preserve the block-wise parallel refinement dynamics.

---

**Algorithm 2** Local branching via partial remasking.

---

**Require:** Trajectory state $\mathbf{z}_t$; prompt $c$; step $t$.
**Ensure:** Expanded state $\mathbf{z}_t^{\mathrm{exp}}$.
1: $\hat{\mathbf{z}}_0 \leftarrow \mathcal{C}_\theta(\mathbf{z}_t, c, t)$.
2: Compute token-wise uncertainty from $\tilde{p}_\theta(\mathbf{z}_0 \mid \mathbf{z}_t, c, t)$ (*e.g.*, entropy).
3: Identify a low-confidence pool $U_t \subseteq \{1, \ldots, L\}$ from the uncertainty scores.
4: Sample a remask subset $I_t \subseteq U_t$ randomly.
5: $\mathbf{z}_t^{\mathrm{exp}} \leftarrow \mathrm{Remask}(\mathbf{z}_t; I_t)$.
6: **return** $\mathbf{z}_t^{\mathrm{exp}}$.

---

We present an overview of PRISM in Fig. 2 and give a detailed introduction to each framework in the following sections.

### 3.2. Self-Verified Feedback (SVF)

Test-time scaling requires a signal for ranking intermediate hypotheses. External verifiers (*e.g.*, separate reward models) incur additional memory and system complexity. We instead reuse the same dLLM as a binary verifier by prompting it to judge the correctness of a completed hypothesis. Concretely, for each trajectory state $\mathbf{z}_t^{(i)}$ we first obtain $\hat{\mathbf{z}}_0^{(i)} = \mathcal{C}_\theta(\mathbf{z}_t^{(i)}, c, t)$, then construct a verification prompt $\pi(c, \hat{\mathbf{z}}_0^{(i)})$ that asks the model to answer `Yes` or `No` only. Let $\ell_\theta(\cdot \mid \pi)$ denote the verifier's logits under prompt $\pi(c, \hat{\mathbf{z}}_0^{(i)})$, we aggregate logits over two small token-ID sets $\mathcal{I}_{\mathtt{Yes}}$ and $\mathcal{I}_{\mathtt{No}}$:

$$
\begin{aligned}
s_{\mathtt{Yes}} &= \max_{y \in \mathcal{I}_{\mathtt{Yes}}} \ell_\theta\left(y \mid \pi(c, \hat{\mathbf{z}}_0^{(i)})\right), \\
s_{\mathtt{No}} &= \max_{n \in \mathcal{I}_{\mathtt{No}}} \ell_\theta\left(n \mid \pi(c, \hat{\mathbf{z}}_0^{(i)})\right).
\end{aligned}
\tag{2}
$$

We define the SVF score as the `Yes` probability under a restricted binary normalization:

$$
\Phi_{\mathrm{SVF}}(\mathbf{z}_t^{(i)}; c) \triangleq \frac{\exp(s_{\mathtt{Yes}})}{\exp(s_{\mathtt{Yes}}) + \exp(s_{\mathtt{No}})}.
\tag{3}
$$

If both scores are undefined, we set $\Phi_{\mathrm{SVF}} = 0.5$.

**Compute accounting and sparse evaluation.** SVF is not free: Eq. (2) and (3) require an additional forward pass (prefill + decoding a single token) per evaluated hypothesis. To maintain efficiency, we (i) apply SVF only after a warm-up period when hypotheses become semantically meaningful, and (ii) evaluate SVF sparsely using a pruning interval $i$. Let $\mathcal{T}_{\mathrm{svf}} \subseteq \{1, \ldots, T\}$ denote timesteps at which SVF is computed, and $W_t$ denote the number of active trajectories at step $t$ under HTS. The total number of SVF calls is then $\sum_{t \in \mathcal{T}_{\mathrm{svf}}} W_t$. In experiments, we report denoising compute (NFE) and verification compute (SVF calls) separately. Since SVF calls are much fewer than NFE, we focus on NFE as the primary compute budget when comparing baselines.

### 3.3. Hierarchical Trajectory Search (HTS)

A naive linear search allocates $T$ denoising steps to all $N$ trajectories, yielding $O(NT)$ denoising cost. We instead adopt a coarse-to-fine allocation: broad exploration at high noise, progressive thinning as structure emerges, and final refinement on a small survivor set. HTS uses the following schedule:

$$
\begin{cases}
\textbf{Exploration} & T_p < t \leq T, \\
\textbf{Thinning} & T_r < t \leq T_p, \\
\textbf{Refinement} & 1 \leq t \leq T_r,
\end{cases}
\tag{4}
$$

where $T_p = \lceil w_{\max} T \rceil$ and $T_r = \lceil w_{\min} T \rceil$ are determined by the pruning window ratio $W = [w_{\min}, w_{\max}]$, satisfying $1 \leq T_r < T_p \leq T$, and denoising proceeds from $t = T$ to $t = 1$.

**Stage I: Stochastic exploration** $(T_p < t \leq T)$. We sample $N$ initial trajectories and run a short warm-up without aggressive pruning. At high noise, completions $\hat{\mathbf{z}}_0$ are unstable, and SVF is less reliable; thus we prioritize diversity. We keep the active width fixed as $W_t = N$ in this stage.

**Stage II: Progressive thinning** $(T_r < t \leq T_p)$. We maintain an active pool size $W_t$ that decays geometrically as the noise decreases:

$$
W_t = \max\left(\left\lfloor N \cdot d^{-(T_p - t)} \right\rfloor, K\right), \qquad d > 1, \tag{5}
$$

and we choose $T_r$ such that $W_{T_r} = K$. For $t = T_p, T_p - 1, \ldots, T_r + 1$, we allocate computation to produce the *next-step* pool of size $W_{t-1}$: (i) compute SVF scores on the current pool of size $W_t$ (optionally only when $t \in \mathcal{T}_{\mathrm{svf}}$), (ii) select the top-$S$ trajectories as seeds, and (iii) local branch around seeds via partial remasking operation (Sec. 3.4) to obtain $W_{t-1}$ children. Only these $W_{t-1}$ children perform the denoising transition from $t$ to $t-1$. A convenient branch factor is

$$
b_t = \left\lceil \frac{W_{t-1}}{S} \right\rceil, \tag{6}
$$

with truncation to match exactly $W_{t-1}$ children.

**Stage III: Final refinement** $(1 \leq t \leq T_r)$. Once the active width reaches $W_{T_r} = K$, branching ceases. We refine the $K$ surviving trajectories independently down to $t = 1$. To avoid wasting compute on already-determined tokens, we adopt an **efficient sampling** strategy within each block, so the realized number of refinement iterations can be smaller than the nominal $T_r$ steps. Concretely, at each iteration we (i) *commit* any masked position whose maximum predicted probability exceeds a confidence threshold $\tau$, and (ii) *early terminate* the current trajectory once an end-of-answer semantic marker (*e.g.*, \boxed{} for math) is detected, in

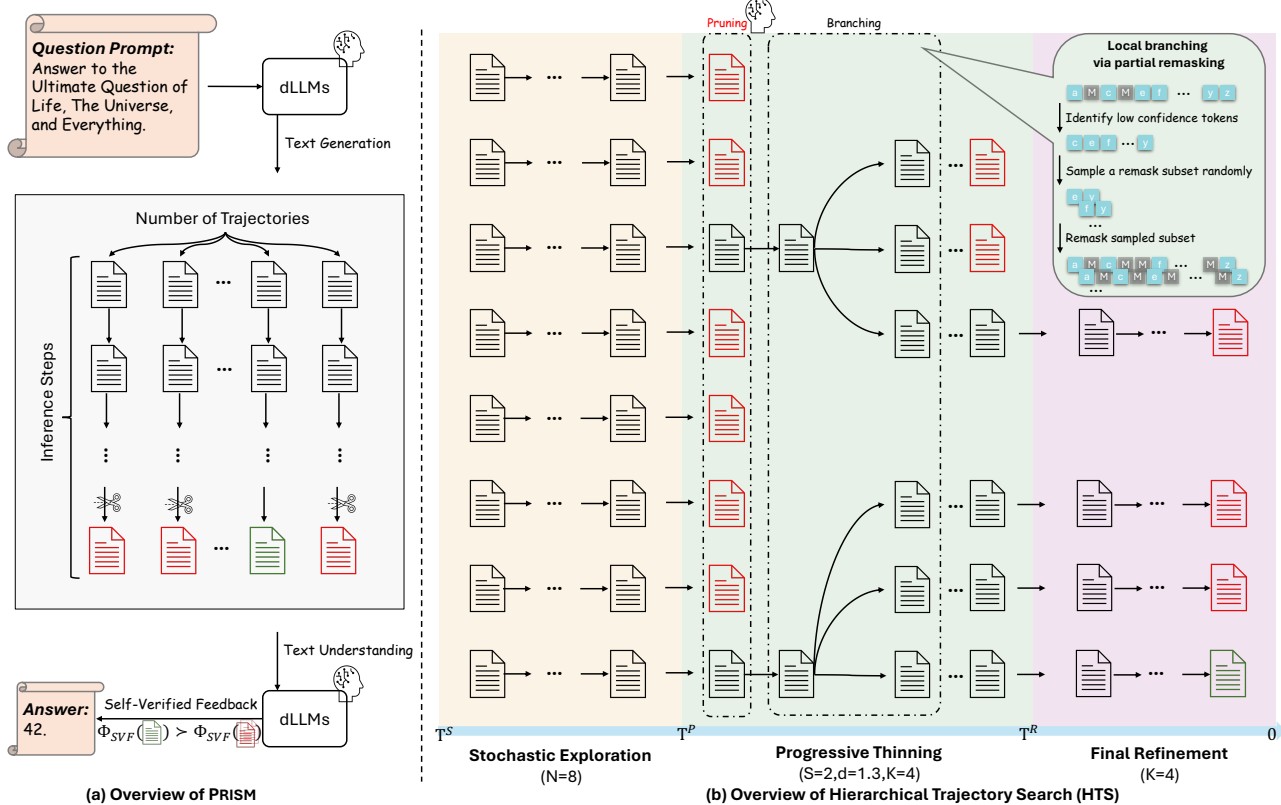

*Figure 2.* Overview of PRISM. (a) Given a prompt, multiple diffusion trajectories are generated in parallel, and intermediate completions are evaluated by Self-Verified Feedback (SVF) using the same dLLM. (b) Hierarchical Trajectory Search (HTS) allocates inference compute dynamically across different stages: stochastic exploration, progressive thinning with SVF-guided pruning and branching, and final refinement on a small survivor set. During thinning, local branching via partial remasking selectively re-masks low-confidence tokens to explore diverse realizations while preserving a high-confidence logic skeleton.

which case the remaining unfilled positions are padded with `eos_id`. Finally, we select the final output using majority voting on the completed samples.

### 3.4. Local Branching via Partial Remasking Operation

To mitigate premature loss of diversity during thinning, we introduce a local branching operator around high-scoring trajectories. Given a survivor state $\mathbf{z}_t$ and its completion $\hat{\mathbf{z}}_0 = \mathcal{C}_\theta(\mathbf{z}_t, c, t)$, we estimate token-wise uncertainty from the $\mathbf{z}_0$-prediction distribution $\tilde{p}_\theta(\mathbf{z}_0 \mid \mathbf{z}_t, c, t)$ (*e.g.*, entropy). We preserve a high-confidence "logic skeleton" and re-mask a complementary subset of low-confidence positions:

$$\mathbf{z}_t = \text{Remask}(\mathbf{z}_t; \mathcal{I}_t), \qquad \mathcal{I}_t \subseteq \{1, \ldots, L\}. \quad (7)$$

Multiple branches are generated by sampling different $\mathcal{I}_t$ per survivor state $\mathbf{z}_t$. Each branch continues denoising from $\mathbf{z}_t$, exploring alternative realizations that remain consistent with the preserved skeleton. Because local branching reuses the current partially specified state instead of restarting from $[m]^L$, it provides targeted diversity while keeping additional denoising cost controlled under a fixed budget.

### 3.5. Algorithm of PRISM

Algorithm 1 summarizes the complete inference pipeline of PRISM, and Algorithm 2 details the local branching operator via partial remasking. Given a prompt $c$, PRISM performs a three-stage Hierarchical Trajectory Search (HTS): (i) stochastic exploration with $N$ trajectories at high noise, (ii) progressive thinning within the pruning window $[T_r, T_p]$ where $T_p = \lceil w_{\max}T \rceil$ and $T_r = \lceil w_{\min}T \rceil$, and (iii) final refinement with width $K$ until completion. During thinning, pruning and branching are executed once every $i$ denoising steps, where SVF ranks trajectories to select top $S$ and each one is expanded using Algorithm 2.

**Complexity analysis.** We measure inference compute by the number of function evaluations (NFE). Algorithm 1 consists of (i) exploration over $N$ trajectories for $T - T_p$ steps, (ii) hierarchical thinning with geometric decay factor $d > 1$, and (iii) final refinement over $K$ trajectories for $T_r$

steps. Therefore, the denoising cost can be written as

$$C_{\text{HTS}} = N(T - T_p) + \sum_{t=T_r+1}^{T_p} |\mathcal{P}_t| + KT_r. \quad (8)$$

In practice, the trajectory pool quickly contracts from $N$ to a smaller set ($K < N$), and the warm-up stage is short ($T - T_p < T$). Hence the overall complexity simplifies to a near-linear scaling:

$$C_{\text{HTS}} \approx O(N + KT), \quad (9)$$

which outperforms conventional linear search baseline with $O(NT)$ complexity.

# 4. Experiments

## 4.1. Experimental Setup

**Tasks.**  We evaluate our method on four reasoning benchmarks spanning two categories: mathematical reasoning and code generation. For mathematical reasoning, we use GSM8K (Cobbe et al., 2021), a benchmark of grade-school arithmetic word problems that requires multi-step symbolic reasoning, and MATH-500 (Hendrycks et al., 2021), a curated set of 500 challenging competition-level mathematics problems. For code generation, we use HumanEval (Chen, 2021), which contains handwritten Python programming problems described in docstrings, MBPP (Austin et al., 2021b), which consists of everyday Python tasks with natural language prompts and associated unit tests.

**Models.**  We leverage three popular dLLMs: LLaDA-8B-Instruct (Nie et al., 2025), Dream-7B-Instruct (Ye et al., 2025), LLaDA-2.0-mini (Bie et al., 2025).

**Baselines.**  We compare against (i) single-trajectory decoding ($N$=1) as the baseline, and (ii) Best-of-$N$ ($N \in 4, 8, 16$), which samples $N$ independent trajectories under identical inference hyperparameters and selects the final output via majority voting.

**Evaluation.**  For all benchmarks, we evaluate models with zero-shot to assess their performance unless otherwise stated. We report accuracy for math reasoning tasks and pass@1 for code generation tasks. All results are reported on the official test sets of each benchmark. We use official checkpoints for all models. To ensure a fair comparison, all baselines are implemented and evaluated under the identical inference setting with the same hyperparameters. To measure computational cost, we adopt the number of function evaluations (NFE) as the metric, consistent with previous studies on inference methods for dLLM (Wu et al., 2025).

## 4.2. Implementation details

**Hyperparameters.**  For math benchmarks (GSM8K and MATH500), we set the generation length to 256 for all models unless otherwise stated. For code benchmarks (HumanEval, MBPP), the generation length is set to 512 across all models unless otherwise stated. For the LLaDA family, we adopt block-autoregressive with a block length of 32 and the number of generation steps is set to 32 for each block unless otherwise stated. We apply low-confidence remasking and set threshold to 0.95 and temperature to 0.7 for all LLaDA-based models. For the Dream family, the number of generation steps is set to the generation length, and we use nucleus sampling with $p = 0.95$ and temperature to 0.1.

**Task Prompts.**  For all evaluation tasks, we use the default prompts provided by `lm-evaluation-harness` v0.4.9.2 (Gao et al., 2024). For self-verification function (SVF), we query with a task-specific prompt that asks for a binary judgment (`Yes`/`No`) on whether the generated solution is likely correct. We present the prompt in Appendix C.

## 4.3. Main Results

The results for mathematical reasoning and code generation are presented in Tab. 1. Across all benchmarks and foundation models, PRISM (K=8) consistently outperforms single-trajectory decoding with at least 26% improvement with a comparable cost to Best-of-4.

**Overall Performance.**  On all three dLLMs, PRISM yields substantial accuracy gains over the $N$=1 baseline. For example, on LLaDA-8B, PRISM (K=8) improves GSM8K accuracy from 67.58% to 85.30% and MATH500 from 26.40% to 42.80%, while also boosting HumanEval and MBPP by 24.39 and 16.40 points, respectively. Similar trends are observed on Dream-7B and LLaDA-2.0-mini, demonstrating the robustness of our method across model scales and paradigms.

**Efficiency-Accuracy Trade-off.**  Compared with linear Best-of-$N$ search, PRISM achieves comparable or better performance with substantially fewer function evaluations. For instance, on LLaDA-8B, PRISM (K=8) reaches 85.30% on GSM8K using 1,048 NFE, whereas Best-of-16 requires 4,096 NFE to achieve 87.50%. This corresponds to over $4\times$ reduction in denoising cost with only marginal accuracy degradation. On MATH500 and MBPP benchmarks, PRISM often matches or surpasses Best-of-16 under less than one-third of the inference budget.

**Effect of Target Width $K$.**  Increasing the final target width $K$ consistently improves performance across tasks. Small values (*e.g.*, $K$=2) already provide noticeable gains over the baseline with minimal overhead, while moderate

*Table 1.* Performance on math and code benchmarks with NFE metrics. We report accuracy on GSM8K and MATH500, and Pass@1 on HumanEval and MBPP. Annotations indicate absolute and relative improvements over single-trajectory decoding ($N$=1), as well as additional SVF calls. For PRISM, we fix the initial width as $N$=16, set the number of survivors to $S = K/2$ for $K \in \{2, 4, 8\}$, and report three target widths $K \in \{2, 4, 8\}$.

| Model | Math | | | | Code | | | |
|---|---|---|---|---|---|---|---|---|
| | **GSM8K** | NFE | **MATH500** | NFE | **HumanEval** | NFE | **MBPP** | NFE |
| LLaDA 8B Instruct | 67.58 | 256 | 26.40 | 256 | 54.88 | 512 | 21.80 | 512 |
| +bst4 | 69.32 | 1024 | 32.00 | 1024 | 77.44 | 2048 | 32.80 | 2048 |
| +bst8 | 82.73 | 2048 | 36.80 | 2048 | 81.71 | 4096 | 33.20 | 4096 |
| +bst16 | 87.50 | 4096 | 38.00 | 4096 | 82.32 | 8192 | 35.20 | 8192 |
| +PRISM (K=2) | 74.24 $^{\Delta+6.66}_{(9.9\%\uparrow)}$ | 283 $^{+27\,(SVF)}_{(298.0\%)}$ | 30.16 $^{\Delta+3.76}_{(14.2\%\uparrow)}$ | 334 $^{+27\,(SVF)}_{(130.5\%)}$ | 71.34 $^{\Delta+16.46}_{(30.0\%\uparrow)}$ | 549 $^{+27\,(SVF)}_{(107.2\%)}$ | 29.40 $^{\Delta+7.60}_{(34.9\%\uparrow)}$ | 561 $^{+27\,(SVF)}_{(109.6\%)}$ |
| +PRISM (K=4) | 75.30 $^{\Delta+7.72}_{(11.4\%\uparrow)}$ | 509 $^{+29\,(SVF)}_{(198.8\%)}$ | 37.70 $^{\Delta+11.30}_{(42.8\%\uparrow)}$ | 622 $^{+29\,(SVF)}_{(243.0\%)}$ | 76.19 $^{\Delta+21.31}_{(38.8\%\uparrow)}$ | 1133 $^{+29\,(SVF)}_{(221.3\%)}$ | 32.40 $^{\Delta+10.60}_{(48.6\%\uparrow)}$ | 1196 $^{+29\,(SVF)}_{(233.6\%)}$ |
| +PRISM (K=8) | 85.30 $^{\Delta+17.72}_{(26.2\%\uparrow)}$ | 1048 $^{+33\,(SVF)}_{(409.4\%)}$ | 42.80 $^{\Delta+16.40}_{(62.1\%\uparrow)}$ | 1304 $^{+33\,(SVF)}_{(509.4\%)}$ | 79.27 $^{\Delta+24.39}_{(44.4\%\uparrow)}$ | 2480 $^{+33\,(SVF)}_{(484.4\%)}$ | 38.20 $^{\Delta+16.40}_{(75.2\%\uparrow)}$ | 2576 $^{+33\,(SVF)}_{(503.1\%)}$ |
| Dream 7B Instruct | 39.09 | 256 | 21.00 | 256 | 42.68 | 512 | 15.60 | 512 |
| +bst4 | 44.55 | 1024 | 25.80 | 1024 | 46.34 | 2048 | 18.40 | 2048 |
| +bst8 | 51.89 | 2048 | 27.80 | 2048 | 47.56 | 4096 | 25.00 | 4096 |
| +bst16 | 55.61 | 4096 | 29.20 | 4096 | 55.49 | 8192 | 25.80 | 8192 |
| +PRISM (K=2) | 40.45 $^{\Delta+1.36}_{(3.5\%\uparrow)}$ | 763 $^{+25\,(SVF)}_{(298.0\%)}$ | 24.80 $^{\Delta+3.80}_{(18.1\%\uparrow)}$ | 876 $^{+25\,(SVF)}_{(342.2\%)}$ | 48.78 $^{\Delta+6.10}_{(14.3\%\uparrow)}$ | 1172 $^{+25\,(SVF)}_{(233.0\%)}$ | 24.00 $^{\Delta+8.40}_{(53.8\%\uparrow)}$ | 1089 $^{+25\,(SVF)}_{(212.7\%)}$ |
| +PRISM (K=4) | 44.24 $^{\Delta+5.15}_{(13.2\%\uparrow)}$ | 852 $^{+27\,(SVF)}_{(332.8\%)}$ | 25.40 $^{\Delta+4.40}_{(21.0\%\uparrow)}$ | 1088 $^{+27\,(SVF)}_{(425.0\%)}$ | 54.88 $^{\Delta+12.20}_{(28.6\%\uparrow)}$ | 1305 $^{+27\,(SVF)}_{(251.6\%)}$ | 26.80 $^{\Delta+11.20}_{(71.8\%\uparrow)}$ | 1175 $^{+27\,(SVF)}_{(229.5\%)}$ |
| +PRISM (K=8) | 53.94 $^{\Delta+14.85}_{(38.0\%\uparrow)}$ | 1076 $^{+30\,(SVF)}_{(420.3\%)}$ | 29.60 $^{\Delta+8.60}_{(41.0\%\uparrow)}$ | 1557 $^{+30\,(SVF)}_{(608.2\%)}$ | 57.32 $^{\Delta+14.64}_{(34.3\%\uparrow)}$ | 1573 $^{+30\,(SVF)}_{(284.2\%)}$ | 30.40 $^{\Delta+14.80}_{(94.9\%\uparrow)}$ | 1294 $^{+30\,(SVF)}_{(252.7\%)}$ |
| LLaDA 2.0 mini | 52.35 | 256 | 20.40 | 256 | 34.76 | 512 | 17.60 | 512 |
| +bst4 | 66.67 | 1024 | 27.00 | 1024 | 75.00 | 2048 | 22.40 | 2048 |
| +bst8 | 74.47 | 2048 | 29.60 | 2048 | 80.49 | 4096 | 23.60 | 4096 |
| +bst16 | 76.89 | 4096 | 30.60 | 4096 | 82.32 | 8192 | 28.80 | 8192 |
| +PRISM (K=2) | 57.73 $^{\Delta+5.38}_{(10.3\%\uparrow)}$ | 325 $^{+27\,(SVF)}_{(127.0\%)}$ | 24.80 $^{\Delta+4.40}_{(21.6\%\uparrow)}$ | 325 $^{+27\,(SVF)}_{(127.0\%)}$ | 50.00 $^{\Delta+15.24}_{(43.8\%\uparrow)}$ | 707 $^{+27\,(SVF)}_{(138.1\%)}$ | 21.00 $^{\Delta+3.40}_{(19.3\%\uparrow)}$ | 704 $^{+27\,(SVF)}_{(137.5\%)}$ |
| +PRISM (K=4) | 66.59 $^{\Delta+14.24}_{(27.2\%\uparrow)}$ | 633 $^{+29\,(SVF)}_{(247.3\%)}$ | 30.00 $^{\Delta+9.60}_{(47.1\%\uparrow)}$ | 650 $^{+29\,(SVF)}_{(253.9\%)}$ | 72.00 $^{\Delta+37.24}_{(107.1\%\uparrow)}$ | 1485 $^{+29\,(SVF)}_{(290.0\%)}$ | 26.80 $^{\Delta+9.20}_{(52.3\%\uparrow)}$ | 1489 $^{+29\,(SVF)}_{(290.8\%)}$ |
| +PRISM (K=8) | 75.91 $^{\Delta+23.56}_{(45.0\%\uparrow)}$ | 2072 $^{+33\,(SVF)}_{(809.4\%)}$ | 32.60 $^{\Delta+12.20}_{(59.8\%\uparrow)}$ | 1336 $^{+33\,(SVF)}_{(521.9\%)}$ | 82.32 $^{\Delta+47.56}_{(136.8\%\uparrow)}$ | 3168 $^{+33\,(SVF)}_{(618.8\%)}$ | 32.20 $^{\Delta+14.60}_{(83.0\%\uparrow)}$ | 3180 $^{+33\,(SVF)}_{(621.1\%)}$ |

values (*e.g.*, $K$=4 and $K$=8) offer the best balance between accuracy and efficiency.

**Impact of Self-Verified Feedback.** The additional SVF calls, reported separately in Tab. 1, remain sparse compared to denoising steps. In most settings, the number of SVF evaluations is less than 10% of the total NFE. This confirms that SVF provides an effective verification signal with negligible computational overhead, enabling adaptive pruning and selection without external reward models.

Overall, these results demonstrate that PRISM can reliably transform additional inference compute into accuracy improvements for dLLMs, while avoiding the prohibitive computation cost of naive width scaling.

**Qualitative Examples.** We present qualitative examples between baselines and PRISM in Appendix D.

### 4.4. SVF Reliability Analysis

SVF is used to rank intermediate completions during the pruning window, where ground-truth final correctness is

*Table 2.* Comparison with ReMDM on TruthfulQA.

| Method | TruthfulQA $\Delta$ ROUGE-1/2/L | Inference Time (s) |
|---|---|---|
| LLaDA | $27.1_{\pm0.4}$ / $30.1_{\pm0.4}$ / $27.2_{\pm0.4}$ | 941.5 |
| LLaDA-ReMDM | $29.5_{\pm0.4}$ / $31.8_{\pm0.4}$ / $29.5_{\pm0.3}$ | 1354.8 |
| PRISM | $31.8_{\pm0.4}/35.5_{\pm0.4}/31.9_{\pm0.4}$ | 1048.0 |

*Table 3.* Comparison with external verifiers on GSM8K.

| Verifier | Pass@1 | Params loaded |
|---|---|---|
| SVF (Ours) | 85.30 | 8B |
| Qwen-7B | 84.39 ↓ | 15B |
| Qwen2-7B | 85.98 ↑ | 15B |
| Qwen3-8B | 87.35 ↑ | 16B |

not directly observable. To evaluate whether SVF provides a meaningful pruning signal, we use Qwen3-235B-A22B-Instruct as a strong proxy evaluator. For each pruning event, the proxy evaluator produces keep/prune reference decisions on intermediate completions, and we compare SVF decisions against these references using F1 and Expected

*Table 4.* Component ablation on GSM8K with LLaDA-8B.

| Method | $K = 2$ | $K = 4$ | $K = 8$ |
|---|---|---|---|
| Full PRISM (SVF + remasking) | 74.24 | 75.30 | 85.30 |
| Random pruning (w/o SVF) | 70.68 | 70.98 | 81.59 |
| No remasking | 73.11 | 73.41 | 84.09 |
| Neither SVF nor remasking | 69.09 | 69.39 | 82.12 |

*Table 5.* Reliability of SVF on intermediate completions. We report F1 and ECE at the first and last SVF evaluations within the pruning window. SVF remains informative across both GSM8K and MATH-500, and ECE consistently improves as denoising progresses.

| Dataset | Eval. point | F1, $K = 2/4/8$ | | | ECE, $K = 2/4/8$ | | |
|---|---|---|---|---|---|---|---|
| | | $K = 2$ | $K = 4$ | $K = 8$ | $K = 2$ | $K = 4$ | $K = 8$ |
| GSM8K | First SVF | .676 | .696 | .731 | .330 | .281 | .231 |
| GSM8K | Last SVF | .649 | .631 | .639 | .286 | .170 | .091 |
| MATH-500 | First SVF | .650 | .697 | .704 | .306 | .267 | .204 |
| MATH-500 | Last SVF | .634 | .629 | .631 | .258 | .158 | .097 |

Calibration Error (ECE). Because both SVF and the proxy reference retain a fixed number of survivors at each pruning event, precision and recall are numerically identical; we therefore report F1.

As shown in Table 5, SVF achieves F1 scores between $0.63$ and $0.73$, indicating that it provides a useful ranking signal beyond random pruning. More importantly, ECE decreases from the first to the last SVF evaluation across all $K$ values on both benchmarks. This trend suggests that SVF confidence becomes better calibrated as denoising progresses and intermediate completions become more semantically stable. The MATH-500 results are close to GSM8K despite the much lower single-trajectory base accuracy on MATH-500, suggesting that SVF does not collapse in harder low-accuracy regimes.

### 4.5. Comparison with Other TTS Methods

We compare PRISM with recent test-time scaling methods (Chen et al., 2025; Huang et al., 2025b; Wang et al., 2025). Since MEDAL and RFG are not open-sourced and their reported results are obtained under different inference settings, we summarize their published performance and compute as reference points. MEDAL (Huang et al., 2025b) reports using $12.3\times$ the baseline runtime and achieving higher accuracy than best-of-15 on GSM8K (66.7 vs. 65.3). Under our setting, PRISM (N=20, S=4, K=8) uses roughly $8.38\times$ the baseline denoising compute (NFE) and achieves better performance than best-of-15 on GSM8K (87.88 vs 86.74). RFG (Chen et al., 2025) reports accuracy improvements of up to 9.2% across four benchmarks with about $2\times$ NFE, whereas PRISM achieves $> 10\%$ gains with a comparable $\sim 2\times$ NFE budget. For ReMDM (Wang et al., 2025), we run a direct head-to-head comparison on TruthfulQA (Lin et al., 2022) using its default hyperparameters

(Tab. 2). Inference time is measured on a single H100 GPU.

Overall, these results suggest that PRISM achieves a better performance-efficiency trade-off, often matching best-of-$N$ performance with substantially fewer function evaluations (NFE).

### 4.6. Comparison with External Verifiers.

We compare SVF with external verifier models of comparable scale. Specifically, we evaluate LLaDA-8B-Instruct with SVF against the same model paired with external LLM-based verifiers, including Qwen-7B (Bai et al., 2023), Qwen2-7B (Team et al., 2024), and Qwen3-8B (Team, 2025), on GSM8K. Tab. 3 reports the results. While external verifiers can yield better performance, they require loading and running a separate model during inference, substantially increasing memory usage and often exceeding the capacity of a 40GB A100. In contrast, SVF is designed to enable efficient test-time scaling without introducing extra models which would double deployment memory.

### 4.7. Component Ablation

Table 4 isolates the contributions of SVF and partial remasking. Replacing SVF-guided pruning with random pruning reduces accuracy by 3.56, 4.32, and 3.71 points for $K = 2, 4, 8$, respectively, showing that SVF is the main source of adaptive search gains. Removing partial remasking causes a smaller but consistent drop of 1.13–1.89 points, indicating that local branching improves diversity without restarting trajectories from scratch. Removing both mechanisms yields the largest degradation, confirming that guided pruning and local remasking are complementary.

### 4.8. Hyperparameter Analysis

We study the sensitivity of PRISM to key hyperparameters in Hierarchical Trajectory Search (HTS) and Self-Verified Feedback (SVF) and present detailed analyses on HumanEval, GSM8K, Math-500 and MBPP using LLaDA 8B Instruct under the same inference setup in Appendix B.

## 5. Conclusion

We present PRISM, a framework that unlocks efficient test-time scaling for discrete diffusion language models. We designed a hierarchical search algorithm that concentrates compute on the critical early-to-mid denoising window. PRISM demonstrates that dLLMs can achieve competitive mathematical reasoning and code generation performance with significantly lower inference cost than vanilla width-scaling methods, paving the way for non-autoregressive models to serve as powerful reasoners.

**Acknowledgement.** The authors are grateful for partial supports by NSF Grant DMS-2513699 (Yuchen Zhu & Molei Tao), DOE Grants NA0004261 (Molei Tao), SC0026274 (Yuchen Zhu & Molei Tao), Richard Duke Fellowship (Yuchen Zhu & Molei Tao), and Simons Institute for the Theory of Computing at UC Berkeley (Molei Tao), National Key Research and Development Program of China No. 2021ZD0201300 (Jianru Xue).

## Impact Statement

This paper proposes an efficient test-time scaling framework for discrete diffusion language models, aiming to improve reasoning and generation quality under a constrained inference budget. By reallocating computation via hierarchical search and replacing external verifiers with lightweight self-verification, our approach can reduce additional memory overhead and improve the accessibility of test-time scaling.

Potential risks are similar to those of general-purpose language models: stronger inference-time reasoning could be misused to generate harmful or misleading content, and self-verification may be imperfect or overconfident on out-of-distribution inputs. Our method does not introduce new data collection or user profiling, and it inherits the biases and limitations of the underlying pretrained models.

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

# A. Entropy Analysis

We provide an auxiliary diagnostic on the uncertainty dynamics of DREAM-7B-INSTRUCT, one of the dLLMs evaluated in our main experiments. Specifically, we track the token-averaged predictive entropy along the denoising trajectory. For each benchmark (GSM8K, HumanEval, Math-500, and MBPP), we randomly sample eight independent stochastic trajectories (*e.g.*, different random seeds under the same sampling hyperparameters) and visualize their entropy curves in Figs. 3–6. This analysis complements our NFE-based cost reporting by revealing how quickly the model's distribution sharpens over timesteps and how much trajectory-to-trajectory variability remains throughout decoding. These dynamics also motivate our design of a pruning window: pruning is most effective when applied after the early high-entropy phase, where the model's uncertainty has substantially decreased while multiple plausible trajectories still coexist. In the plots, we highlight eight final trajectories (colored); the light gray trajectories correspond to branches that are pruned during progressive thinning stage.

**Token-averaged predictive entropy.** At each timestep $t$, the model produces a categorical distribution over the vocabulary for every token position. We compute the entropy per position and then average over the $L$ positions:

$$\mathcal{H}(t) = \frac{1}{L} \sum_{i=1}^{L} H\left(p_\theta(\cdot \mid \mathbf{z}_t, c, t)_i\right), \qquad H(p) = -\sum_{v \in \mathcal{V}} p(v) \log p(v), \tag{10}$$

where $p_\theta(\cdot \mid \mathbf{z}_t, c, t)_i$ denotes the predicted token distribution at position $i$ conditioned on the current noisy state $\mathbf{z}_t$, the prompt/context $c$, and timestep $t$ (we use the natural logarithm). Intuitively, $\mathcal{H}(t)$ summarizes the model's average uncertainty about token identities at timestep $t$; lower entropy indicates a sharper, more confident predictive distribution.

**Qualitative observations.** Across all four benchmarks, entropy drops sharply in the very early timesteps and then decays more gradually, with occasional non-monotonic "bumps" that reflect stochastic exploration and local ambiguity. We also observe larger trajectory-to-trajectory variance on code generation benchmarks than on GSM8K, suggesting that early-to-mid decoding can sustain multiple plausible partial programs before converging near completion.

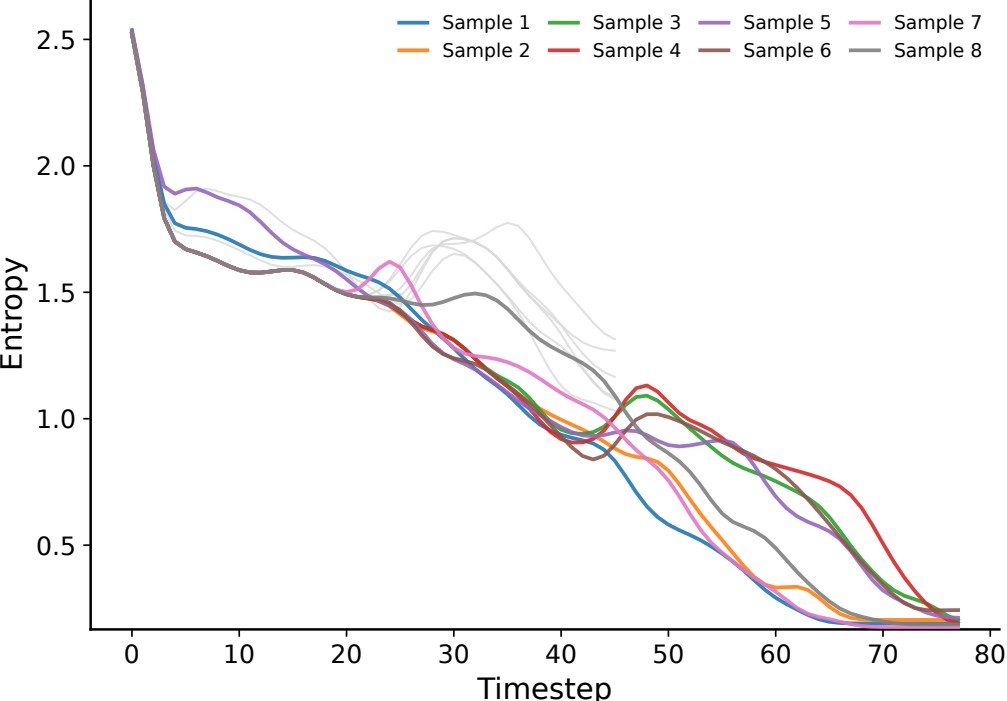

*Figure 3.* Token-averaged predictive entropy trajectories of DREAM-7B-INSTRUCT on GSM8K. Each curve corresponds to one independently sampled decoding trajectory under identical inference settings, and the y-axis reports $\mathcal{H}(t)$ from Eq. (10) (entropy averaged over all token positions at each timestep). The light gray curves indicate trajectories that are pruned during thinning (shown only up to the timestep where they are discarded). Entropy decreases rapidly at the beginning, followed by a smoother decay with mild mid-trajectory fluctuations, and all runs converge to a low-entropy regime near the end of decoding, indicating increasing confidence in token identities as denoising progresses.

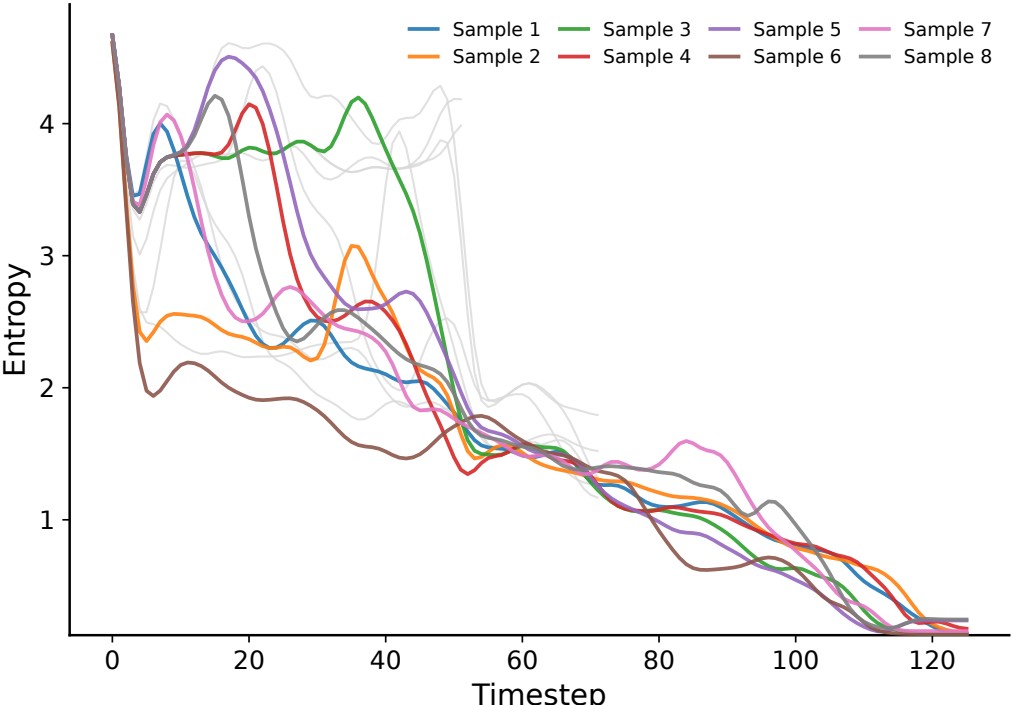

*Figure 4.* Token-averaged predictive entropy trajectories of DREAM-7B-INSTRUCT on HumanEval. Each curve corresponds to one independently sampled decoding trajectory under identical inference settings, and the y-axis reports $\mathcal{H}(t)$ from Eq. (10) (entropy averaged over all token positions at each timestep). The light gray curves indicate trajectories that are pruned during thinning (shown only up to the timestep where they are discarded). Compared with GSM8K, the curves exhibit a more pronounced high-entropy plateau and larger inter-trajectory variance in the early-to-mid timesteps, consistent with multiple competing program structures remaining plausible for longer. Despite such variability, all trajectories eventually enter a low-entropy phase and converge toward completion.

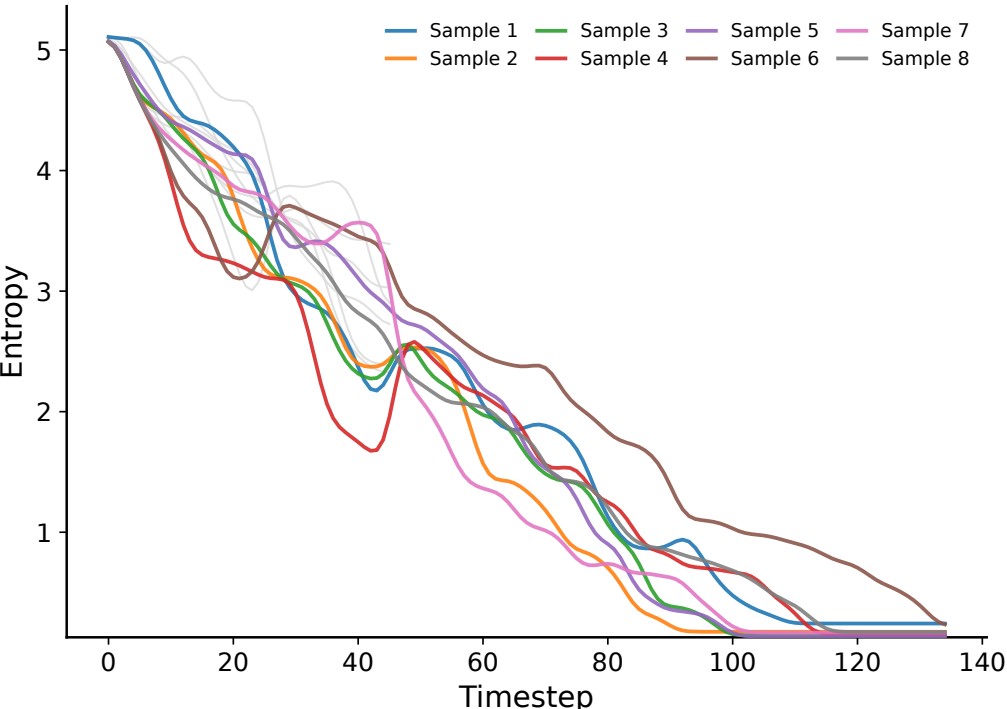

*Figure 5.* Token-averaged predictive entropy trajectories of DREAM-7B-INSTRUCT on Math-500. Each curve corresponds to one independently sampled decoding trajectory under identical inference settings, and the y-axis reports $\mathcal{H}(t)$ from Eq. (10) (entropy averaged over all token positions at each timestep). The light gray curves indicate trajectories that are pruned during thinning (shown only up to the timestep where they are discarded). The entropy starts at a relatively high value and decays over a longer horizon, with noticeable differences in decay rate across trajectories, reflecting heterogeneous levels of difficulty and ambiguity during mathematical reasoning. Near late timesteps, trajectories progressively collapse to low entropy as predictions become more deterministic.

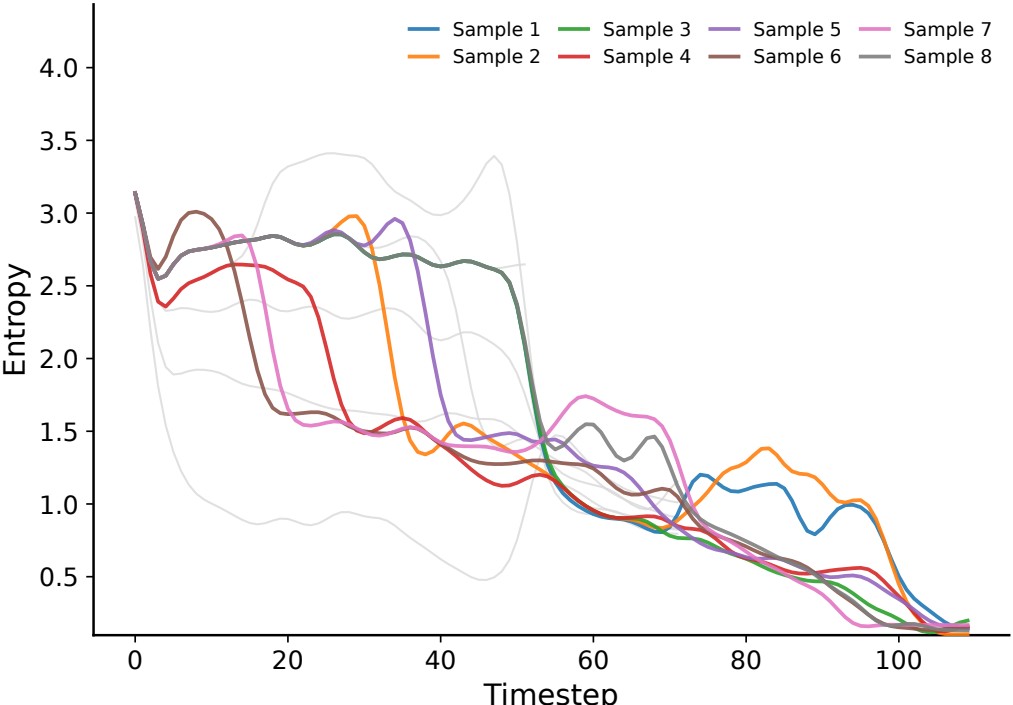

*Figure 6.* Token-averaged predictive entropy trajectories of DREAM-7B-INSTRUCT on MBPP. Each curve corresponds to one independently sampled decoding trajectory under identical inference settings, and the y-axis reports $\mathcal{H}(t)$ from Eq. (10) (entropy averaged over all token positions at each timestep). The light gray curves indicate trajectories that are pruned during thinning (shown only up to the timestep where they are discarded). Similar to HumanEval, MBPP shows substantial trajectory-to-trajectory variability and non-monotonic segments in the mid timesteps, suggesting that the model may maintain multiple plausible partial solutions before committing. All runs nevertheless converge to a low-entropy regime toward the end, indicating increased confidence as denoising completes.

## B. Hyperparameter Analysis

We study the sensitivity of PRISM to key hyperparameters in Hierarchical Trajectory Search (HTS) and Self-Verified Feedback (SVF). All analyses are conducted on HumanEval, GSM8K, Math-500 and MBPP using LLaDA 8B Instruct under the same inference setup. We report task performance (Pass@1 for code and accuracy for math) together with inference cost measured by the number of function evaluations (NFE), and focus our main analysis on HumanEval. For reference, we include a single-trajectory baseline ($N=1$) and a linear width-scaling baseline (Best-of-16). Throughout this section, **Speedup** is computed with respect to Linear Search ($N=16$), *i.e.*, Speedup = $NFE_{linear}/NFE$. We also visualize the hyperparameter combinations across the four benchmarks in Fig. 7, 8, 9, and 10.

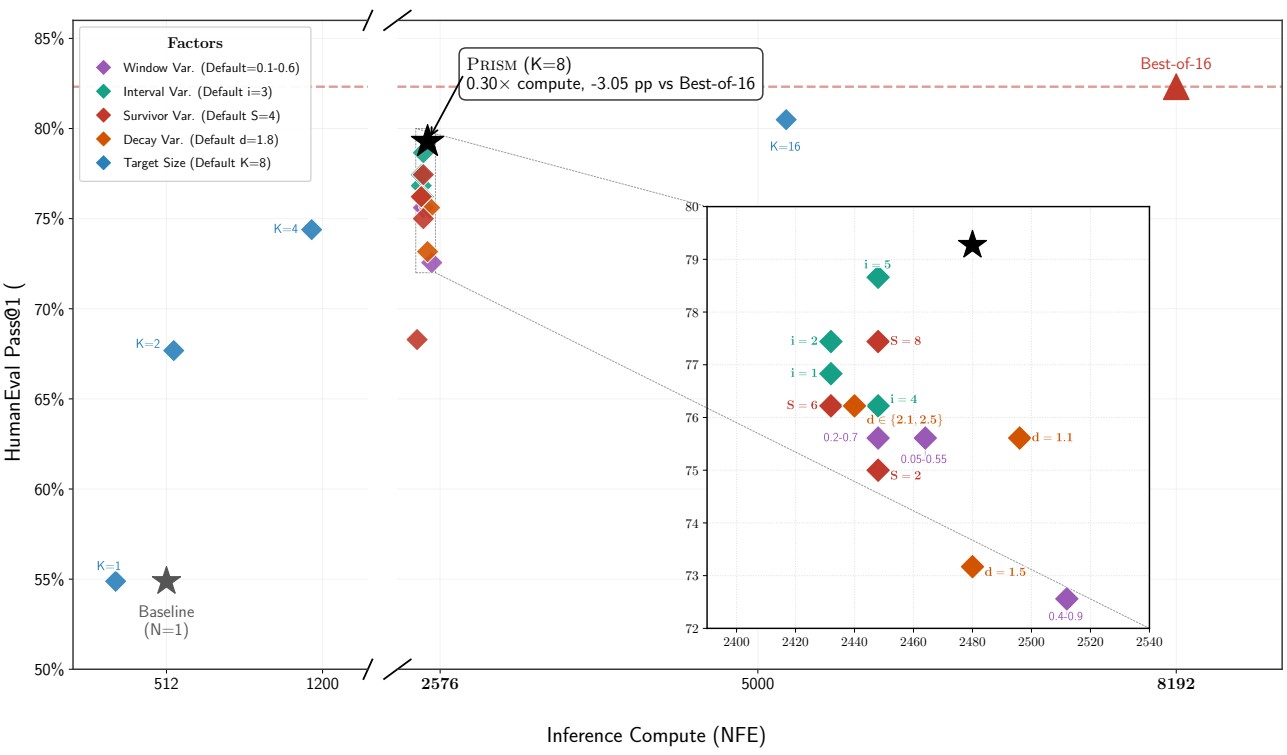

*Figure 7.* PRISM strategy trade-off between HumanEval Pass@1 and inference compute (NFE).

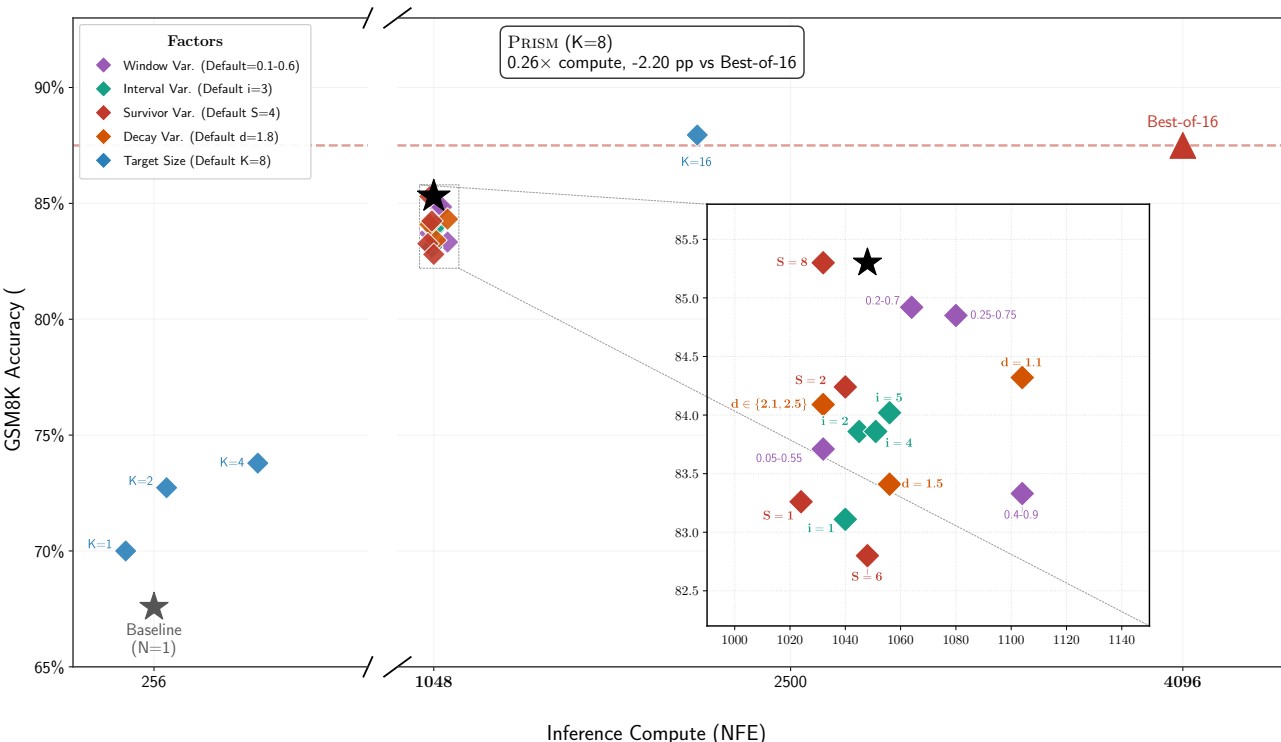

Figure 8. PRISM strategy trade-off between GSM8K Accuracy and inference compute (NFE).

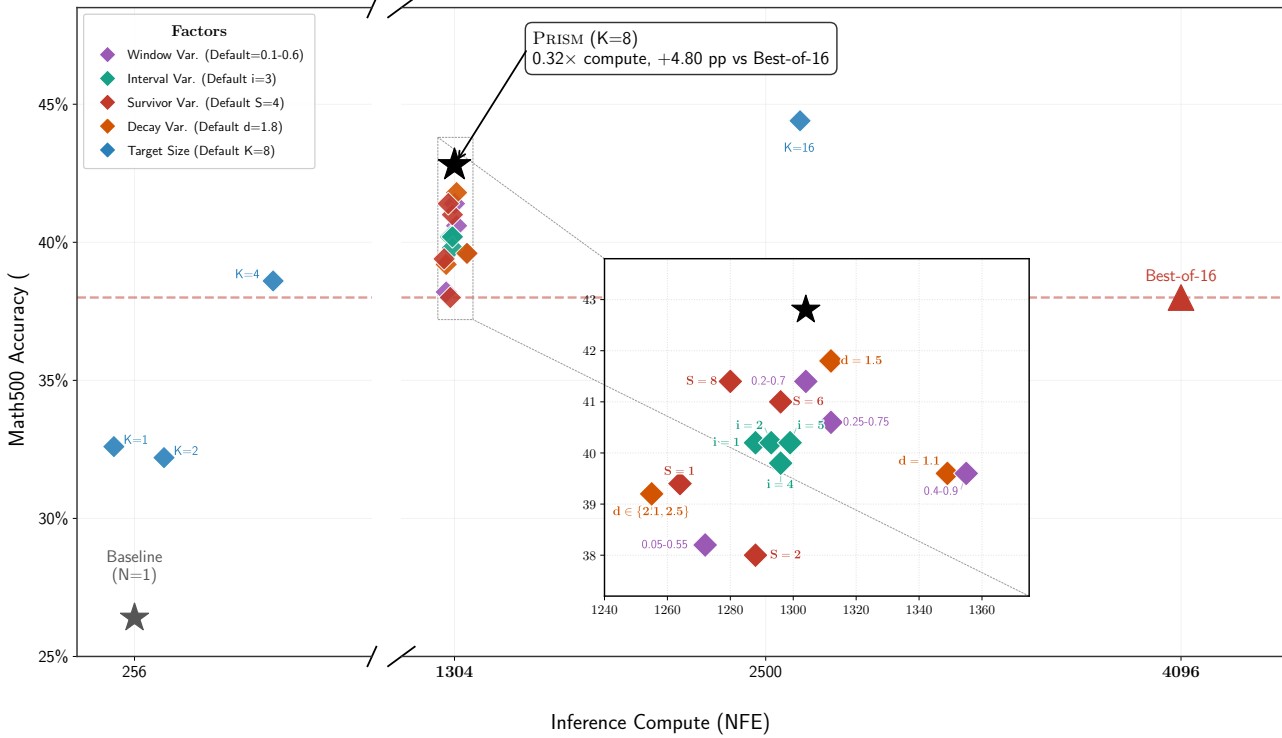

Figure 9. PRISM strategy trade-off between Math500 Accuracy and inference compute (NFE).

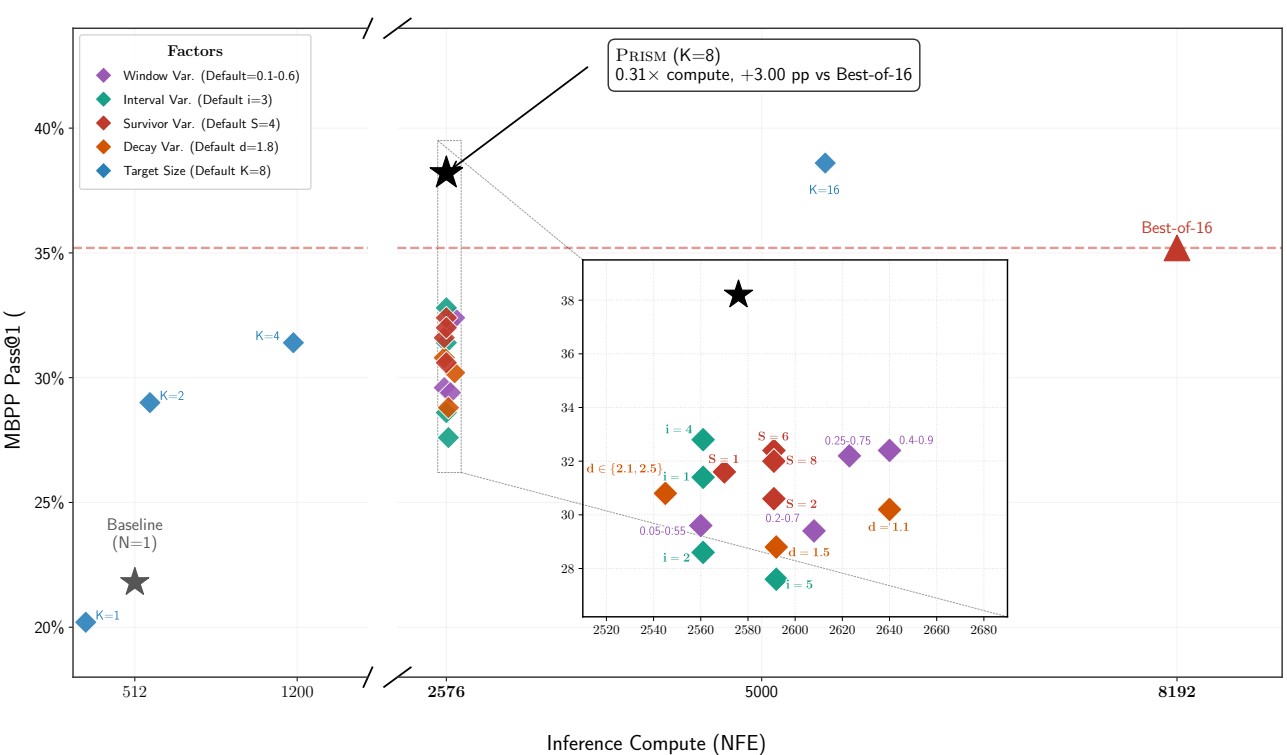

*Figure 10.* PRISM strategy trade-off between MBPP Pass@1 and inference compute (NFE).

## B.1. Analyses on HumanEval

**Effect of Pruning Window.** Tab. 6 analyzes the pruning window $W = [t_{\min}, t_{\max}]$ (normalized by the expected inference steps $T$), where SVF-guided pruning and branching are activated. We observe a clear sweet spot around $W = 0.1$–$0.6$, which achieves the best Pass@1 (79.27%) among all PRISM configurations in Tab. 6. Pruning too early or too late consistently degrades performance, suggesting that effective compute reallocation should focus on the Logic Phase Transition where the high-level solution skeleton is largely determined.

**Effect of Pruning Interval.** Tab. 7 analyzes the pruning interval $i$, *i.e.*, pruning once every $i$ inference steps within the window. A moderate interval ($i = 3$) performs best, whereas overly frequent pruning (small $i$) can prematurely discard promising trajectories, and overly sparse pruning (large $i$) reduces the benefits of adaptive compute reallocation.

**Effect of Decay.** Tab. 8 analyzes the decay factor $d$ controlling how fast the active trajectory width shrinks during progressive thinning. An intermediate decay ($d = 1.8$) yields the strongest results. Both weaker decay (slower thinning) and stronger decay (more aggressive thinning) lead to noticeable drops in Pass@1.

**Effect of Survivors.** Tab. 9 analyzes the survivor width $S$, *i.e.*, the number of top-ranked trajectories retained at each pruning step before branching. Too small $S$ harms diversity and leads to inferior performance, while too large $S$ dilutes the focus of branching.

**Effect of Final Target.** Tab. 10 analyzes the final target width $K$ used in the refinement stage. Increasing $K$ improves Pass@1 monotonically but comes with a predictable NFE increase. In particular, $K = 8$ achieves a strong efficiency–accuracy trade-off (79.27% at $3.3\times$ speedup).

*Table 6.* Effect of **Pruning Window** on HumanEval (Fixed: $d = 1.8, i = 3, S = 4, K = 8$)

| Method | Window | Decay ($d$) | Intv. ($i$) | Surv. ($S$) | Target ($K$) | NFE | Pass@1 (%) | Speedup |
|---|---|---|---|---|---|---|---|---|
| Baseline (N=1) | – | – | – | – | 1 | 512 | 54.88 | - |
| Linear Search (N=16) | – | – | – | – | 16 | 8192 | 82.32 | 1.00× |
| PRISM (Ours) | 0.05 − 0.55 | 1.8 | 3 | 4 | 8 | 2464 | 75.61 | 3.33× |
| **PRISM (Ours)** | **0.1 − 0.6** | **1.8** | **3** | **4** | **8** | **2480** | **79.27** | **3.30×** |
| PRISM (Ours) | 0.2 − 0.7 | 1.8 | 3 | 4 | 8 | 2448 | 75.61 | 3.35× |
| PRISM (Ours) | 0.25 − 0.75 | 1.8 | 3 | 4 | 8 | 2480 | 72.56 | 3.30× |
| PRISM (Ours) | 0.4 − 0.9 | 1.8 | 3 | 4 | 8 | 2512 | 72.56 | 3.26× |

*Table 7.* Effect of **Pruning Interval** on HumanEval (Fixed: $W = 0.1 − 0.6, d = 1.8, S = 4, K = 8$)

| Method | Intv. ($i$) | Window | Decay ($d$) | Surv. ($S$) | Target ($K$) | NFE | Pass@1 (%) | Speedup |
|---|---|---|---|---|---|---|---|---|
| Baseline (N=1) | – | – | – | – | 1 | 512 | 54.88 | - |
| Linear Search (N=16) | – | – | – | – | 16 | 8192 | 82.32 | 1.00× |
| PRISM (Ours) | $i = 1$ | 0.1 − 0.6 | 1.8 | 4 | 8 | 2432 | 76.83 | 3.37× |
| PRISM (Ours) | $i = 2$ | 0.1 − 0.6 | 1.8 | 4 | 8 | 2432 | 77.44 | 3.37× |
| **PRISM (Ours)** | $i = 3$ | **0.1 − 0.6** | **1.8** | **4** | **8** | **2480** | **79.27** | **3.30×** |
| PRISM (Ours) | $i = 4$ | 0.1 − 0.6 | 1.8 | 4 | 8 | 2448 | 76.22 | 3.35× |
| PRISM (Ours) | $i = 5$ | 0.1 − 0.6 | 1.8 | 4 | 8 | 2448 | 78.66 | 3.35× |

*Table 8.* Effect of **Decay Factor** on HumanEval (Fixed: $W = 0.1 - 0.6, i = 3, S = 4, K = 8$)

| Method | Decay ($d$) | Window | Intv. ($i$) | Surv. ($S$) | Target ($K$) | NFE | Pass@1 (%) | Speedup |
|---|---|---|---|---|---|---|---|---|
| Baseline (N=1) | – | – | – | – | 1 | 512 | 54.88 | - |
| Linear Search (N=16) | – | – | – | – | 16 | 8192 | 82.32 | 1.00× |
| PRISM (Ours) | $d = 1.1$ | 0.1 − 0.6 | 3 | 4 | 8 | 2496 | 75.61 | 3.28× |
| PRISM (Ours) | $d = 1.5$ | 0.1 − 0.6 | 3 | 4 | 8 | 2480 | 73.17 | 3.30× |
| **PRISM (Ours)** | $d = 1.8$ | **0.1 – 0.6** | **3** | **4** | **8** | **2480** | **79.27** | **3.30×** |
| PRISM (Ours) | $d = 2.1$ | 0.1 − 0.6 | 3 | 4 | 8 | 2432 | 76.22 | 3.37× |
| PRISM (Ours) | $d = 2.5$ | 0.1 − 0.6 | 3 | 4 | 8 | 2432 | 76.22 | 3.37× |

*Table 9.* Effect of **Survivors** on HumanEval (Fixed: $W = 0.1 - 0.6, d = 1.8, i = 3, K = 8$)

| Method | Surv. ($S$) | Window | Decay ($d$) | Intv. ($i$) | Target ($K$) | NFE | Pass@1 (%) | Speedup |
|---|---|---|---|---|---|---|---|---|
| Baseline (N=1) | – | – | – | – | 1 | 512 | 54.88 | - |
| Linear Search (N=16) | – | – | – | – | 16 | 8192 | 82.32 | 1.00× |
| PRISM (Ours) | $S = 1$ | 0.1 − 0.6 | 1.8 | 3 | 8 | 2400 | 68.29 | 3.41× |
| PRISM (Ours) | $S = 2$ | 0.1 − 0.6 | 1.8 | 3 | 8 | 2448 | 75.00 | 3.35× |
| **PRISM (Ours)** | $S = 4$ | **0.1 – 0.6** | **1.8** | **3** | **8** | **2480** | **79.27** | **3.30×** |
| PRISM (Ours) | $S = 6$ | 0.1 − 0.6 | 1.8 | 3 | 8 | 2432 | 76.22 | 3.37× |
| PRISM (Ours) | $S = 8$ | 0.1 − 0.6 | 1.8 | 3 | 8 | 2448 | 77.44 | 3.35× |

*Table 10.* Effect of **Final Target Width** on HumanEval (Fixed: $W = 0.1 - 0.6, d = 1.8, i = 3, S = 4$)

| Method | Target ($K$) | Window | Decay ($d$) | Intv. ($i$) | Surv. ($S$) | NFE | Pass@1 (%) | Speedup |
|---|---|---|---|---|---|---|---|---|
| Baseline (N=1) | – | – | – | – | – | 512 | 54.88 | - |
| Linear Search (N=16) | – | – | – | – | – | 8192 | 82.32 | 1.00× |
| PRISM (Ours) | $K = 1$ | 0.1 − 0.6 | 1.8 | 3 | 4 | 288 | 54.88 | 28.44× |
| PRISM (Ours) | $K = 2$ | 0.1 − 0.6 | 1.8 | 3 | 4 | 544 | 67.68 | 15.06× |
| PRISM (Ours) | $K = 4$ | 0.1 − 0.6 | 1.8 | 3 | 4 | 1152 | 74.39 | 7.11× |
| PRISM (Ours) | $K = 8$ | 0.1 − 0.6 | 1.8 | 3 | 4 | 2480 | 79.27 | 3.30× |
| **PRISM (Ours)** | $K = 16$ | **0.1 – 0.6** | **1.8** | **3** | **4** | **5216** | **80.49** | **1.57×** |

## B.2. Analyses on GSM8K

*Table 11.* Effect of **Pruning Window** on GSM8K (Fixed: $d = 1.8, i = 3, S = 4, K = 8$)

| Method | Window | Decay ($d$) | Intv. ($i$) | Surv. ($S$) | Target ($K$) | NFE | Pass@1 (%) | Speedup |
|---|---|---|---|---|---|---|---|---|
| Baseline (N=1) | – | – | – | – | 1 | 256 | 67.58 | - |
| Linear Search (N=16) | – | – | – | – | 16 | 4096 | 87.50 | 1.00× |
| PRISM (Ours) | 0.05 − 0.55 | 1.8 | 3 | 4 | 8 | 1032 | 83.71 | 3.97× |
| **PRISM (Ours)** | **0.1 − 0.6** | **1.8** | **3** | **4** | **8** | **1048** | **85.30** | **3.91×** |
| PRISM (Ours) | 0.2 − 0.7 | 1.8 | 3 | 4 | 8 | 1064 | 84.92 | 3.85× |
| PRISM (Ours) | 0.25 − 0.75 | 1.8 | 3 | 4 | 8 | 1080 | 84.85 | 3.79× |
| PRISM (Ours) | 0.4 − 0.9 | 1.8 | 3 | 4 | 8 | 1104 | 83.33 | 3.71× |

*Table 12.* Effect of **Pruning Interval** on GSM8K (Fixed: $W = 0.1 − 0.6, d = 1.8, S = 4, K = 8$)

| Method | Intv. ($i$) | Window | Decay ($d$) | Surv. ($S$) | Target ($K$) | NFE | Pass@1 (%) | Speedup |
|---|---|---|---|---|---|---|---|---|
| Baseline (N=1) | – | – | – | – | 1 | 256 | 67.58 | - |
| Linear Search (N=16) | – | – | – | – | 16 | 4096 | 87.50 | 1.00× |
| PRISM (Ours) | $i = 1$ | 0.1 − 0.6 | 1.8 | 4 | 8 | 1040 | 83.11 | 3.94× |
| PRISM (Ours) | $i = 2$ | 0.1 − 0.6 | 1.8 | 4 | 8 | 1048 | 83.86 | 3.91× |
| **PRISM (Ours)** | **$i = 3$** | **0.1 − 0.6** | **1.8** | **4** | **8** | **1048** | **85.30** | **3.91×** |
| PRISM (Ours) | $i = 4$ | 0.1 − 0.6 | 1.8 | 4 | 8 | 1048 | 83.86 | 3.91× |
| PRISM (Ours) | $i = 5$ | 0.1 − 0.6 | 1.8 | 4 | 8 | 1056 | 84.02 | 3.88× |

*Table 13.* Effect of **Decay Factor** on GSM8K (Fixed: $W = 0.1 − 0.6, i = 3, S = 4, K = 8$)

| Method | Decay ($d$) | Window | Intv. ($i$) | Surv. ($S$) | Target ($K$) | NFE | Pass@1 (%) | Speedup |
|---|---|---|---|---|---|---|---|---|
| Baseline (N=1) | – | – | – | – | 1 | 256 | 67.58 | - |
| Linear Search (N=16) | – | – | – | – | 16 | 4096 | 87.50 | 1.00× |
| PRISM (Ours) | $d = 1.1$ | 0.1 − 0.6 | 3 | 4 | 8 | 1104 | 84.32 | 3.71× |
| PRISM (Ours) | $d = 1.5$ | 0.1 − 0.6 | 3 | 4 | 8 | 1056 | 83.41 | 3.88× |
| **PRISM (Ours)** | **$d = 1.8$** | **0.1 − 0.6** | **3** | **4** | **8** | **1048** | **85.30** | **3.91×** |
| PRISM (Ours) | $d = 2.1$ | 0.1 − 0.6 | 3 | 4 | 8 | 1032 | 84.09 | 3.97× |
| PRISM (Ours) | $d = 2.5$ | 0.1 − 0.6 | 3 | 4 | 8 | 1032 | 84.09 | 3.97× |

*Table 14.* Effect of **Survivors** on GSM8K (Fixed: $W = 0.1 - 0.6, d = 1.8, i = 3, K = 8$)

| Method | Surv. ($S$) | Window | Decay ($d$) | Intv. ($i$) | Target ($K$) | NFE | Pass@1 (%) | Speedup |
|---|---|---|---|---|---|---|---|---|
| Baseline (N=1) | – | – | – | – | 1 | 256 | 67.58 | - |
| Linear Search (N=16) | – | – | – | – | 16 | 4096 | 87.50 | 1.00× |
| PRISM (Ours) | $S = 1$ | 0.1 – 0.6 | 1.8 | 3 | 8 | 1024 | 83.26 | 4.00× |
| PRISM (Ours) | $S = 2$ | 0.1 – 0.6 | 1.8 | 3 | 8 | 1040 | 84.24 | 3.94× |
| **PRISM (Ours)** | $S = 4$ | **0.1 – 0.6** | **1.8** | **3** | **8** | **1048** | **85.30** | **3.91×** |
| PRISM (Ours) | $S = 6$ | 0.1 – 0.6 | 1.8 | 3 | 8 | 1048 | 82.80 | 3.91× |
| PRISM (Ours) | $S = 8$ | 0.1 – 0.6 | 1.8 | 3 | 8 | 1032 | 85.30 | 3.97× |

*Table 15.* Effect of **Final Target Width** on GSM8K (Fixed: $W = 0.1 - 0.6, d = 1.8, i = 3, S = 4$)

| Method | Target ($K$) | Window | Decay ($d$) | Intv. ($i$) | Surv. ($S$) | NFE | Pass@1 (%) | Speedup |
|---|---|---|---|---|---|---|---|---|
| Baseline (N=1) | – | – | – | – | – | 256 | 67.58 | - |
| Linear Search (N=16) | – | – | – | – | – | 4096 | 87.50 | 1.00× |
| PRISM (Ours) | $K = 1$ | 0.1 – 0.6 | 1.8 | 3 | 4 | 184 | 70.00 | 22.26× |
| PRISM (Ours) | $K = 2$ | 0.1 – 0.6 | 1.8 | 3 | 4 | 288 | 72.73 | 14.22× |
| PRISM (Ours) | $K = 4$ | 0.1 – 0.6 | 1.8 | 3 | 4 | 520 | 73.79 | 7.88× |
| PRISM (Ours) | $K = 8$ | 0.1 – 0.6 | 1.8 | 3 | 4 | 1048 | 85.30 | 3.91× |
| **PRISM (Ours)** | $K = 16$ | **0.1 – 0.6** | **1.8** | **3** | **4** | **2120** | **87.95** | **1.93×** |

## B.3. Analyses on Math-500

*Table 16.* Effect of **Pruning Window** on Math500 (Fixed: $d = 1.8, i = 3, S = 4, K = 8$)

| Method | Window | Decay ($d$) | Intv. ($i$) | Surv. ($S$) | Target ($K$) | NFE | Pass@1 (%) | Speedup |
|---|---|---|---|---|---|---|---|---|
| Baseline (N=1) | – | – | – | – | 1 | 256 | 26.40 | - |
| Linear Search (N=16) | – | – | – | – | 16 | 4096 | 38.00 | 1.00× |
| PRISM (Ours) | $0.05 - 0.55$ | 1.8 | 3 | 4 | 8 | 1272 | 38.20 | 3.22× |
| **PRISM (Ours)** | **0.1 – 0.6** | **1.8** | **3** | **4** | **8** | **1304** | **42.80** | **3.14×** |
| PRISM (Ours) | $0.2 - 0.7$ | 1.8 | 3 | 4 | 8 | 1304 | 41.40 | 3.14× |
| PRISM (Ours) | $0.25 - 0.75$ | 1.8 | 3 | 4 | 8 | 1312 | 40.60 | 3.12× |
| PRISM (Ours) | $0.4 - 0.9$ | 1.8 | 3 | 4 | 8 | 1352 | 39.60 | 3.03× |

*Table 17.* Effect of **Pruning Interval** on Math500 (Fixed: $W = 0.1 - 0.6, d = 1.8, S = 4, K = 8$)

| Method | Intv. ($i$) | Window | Decay ($d$) | Surv. ($S$) | Target ($K$) | NFE | Pass@1 (%) | Speedup |
|---|---|---|---|---|---|---|---|---|
| Baseline (N=1) | – | – | – | – | 1 | 256 | 26.40 | - |
| Linear Search (N=16) | – | – | – | – | 16 | 4096 | 38.00 | 1.00× |
| PRISM (Ours) | $i = 1$ | $0.1 - 0.6$ | 1.8 | 4 | 8 | 1288 | 40.20 | 3.18× |
| PRISM (Ours) | $i = 2$ | $0.1 - 0.6$ | 1.8 | 4 | 8 | 1296 | 40.20 | 3.16× |
| **PRISM (Ours)** | $i = 3$ | **0.1 – 0.6** | **1.8** | **4** | **8** | **1304** | **42.80** | **3.14×** |
| PRISM (Ours) | $i = 4$ | $0.1 - 0.6$ | 1.8 | 4 | 8 | 1296 | 39.80 | 3.16× |
| PRISM (Ours) | $i = 5$ | $0.1 - 0.6$ | 1.8 | 4 | 8 | 1296 | 40.20 | 3.16× |

*Table 18.* Effect of **Decay Factor** on Math500 (Fixed: $W = 0.1 - 0.6, i = 3, S = 4, K = 8$)

| Method | Decay ($d$) | Window | Intv. ($i$) | Surv. ($S$) | Target ($K$) | NFE | Pass@1 (%) | Speedup |
|---|---|---|---|---|---|---|---|---|
| Baseline (N=1) | – | – | – | – | 1 | 256 | 26.40 | - |
| Linear Search (N=16) | – | – | – | – | 16 | 4096 | 38.00 | 1.00× |
| PRISM (Ours) | $d = 1.1$ | $0.1 - 0.6$ | 3 | 4 | 8 | 1352 | 39.60 | 3.03× |
| PRISM (Ours) | $d = 1.5$ | $0.1 - 0.6$ | 3 | 4 | 8 | 1312 | 41.80 | 3.12× |
| **PRISM (Ours)** | $d = 1.8$ | **0.1 – 0.6** | **3** | **4** | **8** | **1304** | **42.80** | **3.14×** |
| PRISM (Ours) | $d = 2.1$ | $0.1 - 0.6$ | 3 | 4 | 8 | 1272 | 39.20 | 3.22× |
| PRISM (Ours) | $d = 2.5$ | $0.1 - 0.6$ | 3 | 4 | 8 | 1272 | 39.20 | 3.22× |

*Table 19.* Effect of **Survivors** on Math500 (Fixed: $W = 0.1 - 0.6, d = 1.8, i = 3, K = 8$)

| Method | Surv. ($S$) | Window | Decay ($d$) | Intv. ($i$) | Target ($K$) | NFE | Pass@1 (%) | Speedup |
|---|---|---|---|---|---|---|---|---|
| Baseline (N=1) | – | – | – | – | 1 | 256 | 26.40 | - |
| Linear Search (N=16) | – | – | – | – | 16 | 4096 | 38.00 | $1.00\times$ |
| PRISM (Ours) | $S = 1$ | $0.1 - 0.6$ | 1.8 | 3 | 8 | 1264 | 39.40 | $3.24\times$ |
| PRISM (Ours) | $S = 2$ | $0.1 - 0.6$ | 1.8 | 3 | 8 | 1288 | 38.00 | $3.18\times$ |
| **PRISM (Ours)** | $S = 4$ | **0.1 – 0.6** | **1.8** | **3** | **8** | **1304** | **42.80** | **3.14$\times$** |
| PRISM (Ours) | $S = 6$ | $0.1 - 0.6$ | 1.8 | 3 | 8 | 1296 | 41.00 | $3.16\times$ |
| PRISM (Ours) | $S = 8$ | $0.1 - 0.6$ | 1.8 | 3 | 8 | 1280 | 41.40 | $3.20\times$ |

*Table 20.* Effect of **Final Target Width** on Math500 (Fixed: $W = 0.1 - 0.6, d = 1.8, i = 3, S = 4$)

| Method | Target ($K$) | Window | Decay ($d$) | Intv. ($i$) | Surv. ($S$) | NFE | Pass@1 (%) | Speedup |
|---|---|---|---|---|---|---|---|---|
| Baseline (N=1) | – | – | – | – | – | 256 | 26.40 | - |
| Linear Search (N=16) | – | – | – | – | – | 4096 | 38.00 | $1.00\times$ |
| PRISM (Ours) | $K = 1$ | $0.1 - 0.6$ | 1.8 | 3 | 4 | 200 | 32.60 | $20.48\times$ |
| PRISM (Ours) | $K = 2$ | $0.1 - 0.6$ | 1.8 | 3 | 4 | 336 | 32.20 | $12.19\times$ |
| PRISM (Ours) | $K = 4$ | $0.1 - 0.6$ | 1.8 | 3 | 4 | 632 | 38.60 | $6.48\times$ |
| PRISM (Ours) | $K = 8$ | $0.1 - 0.6$ | 1.8 | 3 | 4 | 1304 | 42.80 | $3.14\times$ |
| **PRISM (Ours)** | $K = 16$ | **0.1 – 0.6** | **1.8** | **3** | **4** | **2632** | **44.40** | **1.56$\times$** |

## B.4. Analyses on MBPP

*Table 21.* Effect of **Pruning Window** on MBPP (Fixed: $d = 1.8, i = 3, S = 4, K = 8$)

| Method | Window | Decay ($d$) | Intv. ($i$) | Surv. ($S$) | Target ($K$) | NFE | Pass@1 (%) | Speedup |
|---|---|---|---|---|---|---|---|---|
| Baseline (N=1) | – | – | – | – | 1 | 512 | 21.80 | - |
| Linear Search (N=16) | – | – | – | – | 16 | 8192 | 35.20 | 1.00× |
| PRISM (Ours) | $0.05 - 0.55$ | 1.8 | 3 | 4 | 8 | 2560 | 29.60 | 3.20× |
| **PRISM (Ours)** | **0.1 – 0.6** | **1.8** | **3** | **4** | **8** | **2576** | **38.20** | **3.18×** |
| PRISM (Ours) | $0.2 - 0.7$ | 1.8 | 3 | 4 | 8 | 2608 | 29.40 | 3.14× |
| PRISM (Ours) | $0.25 - 0.75$ | 1.8 | 3 | 4 | 8 | 2608 | 32.20 | 3.14× |
| PRISM (Ours) | $0.4 - 0.9$ | 1.8 | 3 | 4 | 8 | 2640 | 32.40 | 3.10× |

*Table 22.* Effect of **Pruning Interval** on MBPP (Fixed: $W = 0.1 - 0.6, d = 1.8, S = 4, K = 8$)

| Method | Intv. ($i$) | Window | Decay ($d$) | Surv. ($S$) | Target ($K$) | NFE | Pass@1 (%) | Speedup |
|---|---|---|---|---|---|---|---|---|
| Baseline (N=1) | – | – | – | – | 1 | 512 | 21.80 | - |
| Linear Search (N=16) | – | – | – | – | 16 | 8192 | 35.20 | 1.00× |
| PRISM (Ours) | $i = 1$ | $0.1 - 0.6$ | 1.8 | 4 | 8 | 2576 | 31.40 | 3.18× |
| PRISM (Ours) | $i = 2$ | $0.1 - 0.6$ | 1.8 | 4 | 8 | 2576 | 28.60 | 3.18× |
| **PRISM (Ours)** | **$i = 3$** | **0.1 – 0.6** | **1.8** | **4** | **8** | **2576** | **38.20** | **3.18×** |
| PRISM (Ours) | $i = 4$ | $0.1 - 0.6$ | 1.8 | 4 | 8 | 2576 | 32.80 | 3.18× |
| PRISM (Ours) | $i = 5$ | $0.1 - 0.6$ | 1.8 | 4 | 8 | 2592 | 27.60 | 3.16× |

*Table 23.* Effect of **Decay Factor** on MBPP (Fixed: $W = 0.1 - 0.6, i = 3, S = 4, K = 8$)

| Method | Decay ($d$) | Window | Intv. ($i$) | Surv. ($S$) | Target ($K$) | NFE | Pass@1 (%) | Speedup |
|---|---|---|---|---|---|---|---|---|
| Baseline (N=1) | – | – | – | – | 1 | 512 | 21.80 | - |
| Linear Search (N=16) | – | – | – | – | 16 | 8192 | 35.20 | 1.00× |
| PRISM (Ours) | $d = 1.1$ | $0.1 - 0.6$ | 3 | 4 | 8 | 2640 | 30.20 | 3.10× |
| PRISM (Ours) | $d = 1.5$ | $0.1 - 0.6$ | 3 | 4 | 8 | 2592 | 28.80 | 3.16× |
| **PRISM (Ours)** | **$d = 1.8$** | **0.1 – 0.6** | **3** | **4** | **8** | **2576** | **38.20** | **3.18×** |
| PRISM (Ours) | $d = 2.1$ | $0.1 - 0.6$ | 3 | 4 | 8 | 2560 | 30.80 | 3.20× |
| PRISM (Ours) | $d = 2.5$ | $0.1 - 0.6$ | 3 | 4 | 8 | 2560 | 30.80 | 3.20× |

*Table 24.* Effect of **Survivors** on MBPP (Fixed: $W = 0.1 - 0.6, d = 1.8, i = 3, K = 8$)

| Method | Surv. ($S$) | Window | Decay ($d$) | Intv. ($i$) | Target ($K$) | NFE | Pass@1 (%) | Speedup |
|---|---|---|---|---|---|---|---|---|
| Baseline (N=1) | – | – | – | – | 1 | 512 | 21.80 | - |
| Linear Search (N=16) | – | – | – | – | 16 | 8192 | 35.20 | 1.00× |
| PRISM (Ours) | $S = 1$ | $0.1 - 0.6$ | 1.8 | 3 | 8 | 2560 | 31.60 | 3.20× |
| PRISM (Ours) | $S = 2$ | $0.1 - 0.6$ | 1.8 | 3 | 8 | 2576 | 30.60 | 3.18× |
| **PRISM (Ours)** | $S = 4$ | **0.1 – 0.6** | **1.8** | **3** | **8** | **2576** | **38.20** | **3.18×** |
| PRISM (Ours) | $S = 6$ | $0.1 - 0.6$ | 1.8 | 3 | 8 | 2576 | 32.40 | 3.18× |
| PRISM (Ours) | $S = 8$ | $0.1 - 0.6$ | 1.8 | 3 | 8 | 2576 | 32.00 | 3.18× |

*Table 25.* Effect of **Final Target Width** on MBPP (Fixed: $W = 0.1 - 0.6, d = 1.8, i = 3, S = 4$)

| Method | Target ($K$) | Window | Decay ($d$) | Intv. ($i$) | Surv. ($S$) | NFE | Pass@1 (%) | Speedup |
|---|---|---|---|---|---|---|---|---|
| Baseline (N=1) | – | – | – | – | – | 512 | 21.80 | - |
| Linear Search (N=16) | – | – | – | – | – | 8192 | 35.20 | 1.00× |
| PRISM (Ours) | $K = 1$ | $0.1 - 0.6$ | 1.8 | 3 | 4 | 304 | 20.20 | 26.95× |
| PRISM (Ours) | $K = 2$ | $0.1 - 0.6$ | 1.8 | 3 | 4 | 576 | 29.00 | 14.22× |
| PRISM (Ours) | $K = 4$ | $0.1 - 0.6$ | 1.8 | 3 | 4 | 1184 | 31.40 | 6.92× |
| PRISM (Ours) | $K = 8$ | $0.1 - 0.6$ | 1.8 | 3 | 4 | 2576 | 38.20 | 3.18× |
| **PRISM (Ours)** | $K = 16$ | **0.1 – 0.6** | **1.8** | **3** | **4** | **5488** | **38.60** | **1.49×** |

## C. SVF Prompt Template

We use two templates depending on the task family: a *math-judge* prompt for mathematical reasoning benchmarks and a *code-judge* prompt for code generation benchmarks (Fig. 11 and 12). In both cases, we insert the original problem statement and a truncated model completion into the prompt, and the verifier must output a single word decision.

---

**Math Tasks Instruction Template**

*You are an expert mathematician and competition judge. Your task is to evaluate a proposed mathematical solution for a given problem based on its logical rigor and accuracy.*

**[Math Problem]**
*<PROBLEM>*
**[/Math Problem]**

**[Proposed Mathematical Solution]**
*<COMPLETION (TRUNCATED)>*
**[/Proposed Mathematical Solution]**

**Analysis Steps:**
1. Reasoning Validity: Are the logical steps and mathematical properties applied correctly?
2. Calculation Accuracy: Are the intermediate calculations or algebraic manipulations accurate?
3. Goal Alignment: Does the current reasoning path directly lead toward the final answer required by the problem?

**Conclusion:** Based on your analysis, is this solution path sound and likely to result in the correct final answer? Answer with a single word: **Yes** or **No.**
**Answer:**

---

*Figure 11.* Self-verification prompt template for math tasks. The verifier must output a single-word decision (Yes/No).

**Code Tasks Instruction Template**

*You are an expert programming contest judge. Your task is to evaluate a generated solution for a given problem based on correctness, efficiency, and adherence to constraints.*

**[Problem Statement]**
*<PROBLEM>*
**[/Problem Statement]**

**[Proposed Python Solution]**
**[BEGIN_CODE]**
*<COMPLETION (TRUNCATED)>*
**[END_CODE]**
**[/Proposed Python Solution]**

**Analysis Steps:**
1. Correctness: Does the core algorithm correctly solve the problem?
2. Efficiency: Is the time complexity acceptable for the given constraints?
3. Edge Cases & Constraints: Does the code handle all rules and edge cases?

**Conclusion:** Based on your analysis, is the solution likely to be fully correct?
Answer with a single word: **Yes** or **No**.
**Answer:**

*Figure 12.* Self-verification prompt template for code tasks. The verifier must output a single-word decision (Yes/No).

## D. Qualitative Examples.

We provide qualitative comparisons between the baseline (LLaDA 8B Instruct) and PRISM on four benchmarks on the following pages.

### D.1. GSM8K

#### D.1.1. LLADA 8B INSTRUCT

---

**Query (GSM8K)**

A merchant wants to make a choice of purchase between 2 purchase plans: jewelry worth $5,000 or electronic gadgets worth $8,000. His financial advisor speculates that the jewelry market will go up 2.5% while the electronic gadgets market will rise 1.2% within the same month. If the merchant is looking to maximize profit at the end of this month, how much profit would this be?

---

**Original (LLaDA 8B Instruct)**

To determine the profit at the end of the month, we need to calculate the expected value of the jewelry and electronic gadgets after their respective market increases.

1. Initial value of jewelry: $5,000
2. Expected value of jewelry: $5,000 + ($5,000 \times 0.025) = $5,125$

3. Initial value of gadgets: $8,000
4. Expected value of gadgets: $8,000 + ($8,000 \times 0.012) = $8,096$

**The merchant should choose the purchase plan with the higher final expected value to maximize profit. In this case, the gadgets have a higher expected value of $8,096.**

Therefore, the merchant should choose the electronic gadgets.

> **8,096**

---

**Ours (PRISM)**

To determine the potential profit from each purchase plan, we need to calculate the increase in value for each item.

1. Jewelry:
- Increase in value: 2.5% of $5,000 = $0.025 \times 5000 = 125$
- Profit: $125

2. Electronic Gadgets:
- Increase in value: 1.2% of $8,000 = $0.012 \times 8000 = 96$
- Profit: $96

To maximize profit, the merchant should choose the jewelry, which yields a profit of $125.

> **125**

---

D.1.2. DREAM 7B INSTRUCT

---

**Query (GSM8K)**

Charlie wants to sell beeswax candles. For every pound of beeswax, he can make 10 tapered candles. One pound of beeswax and the wicks cost $10.00 in supplies. If he sells each candle for $2.00 each, what is his net profit if he makes and sells 20 candles?

---

**Original (Dream 7B Instruct)**

One pound of beeswax makes 10 candles, and it costs $10.00 in supplies.

**1. Calculate the cost of supplies for 20 candles:**
- Cost per 10 candles = $10.00
- Cost for 20 candles = $20 \times \$10.00/10 = \$100.00$

**2. Calculate the total earnings from selling 20 candles:**
- Selling price per candle = $2.00
- Total earnings = $20 \times \$2.00 = \$40.00$

**3. Calculate the net profit:**
- Net profit = Earnings - Cost of supplies
- Net profit = $\$40.00 - \$100.00 = \$60.00$

The answer is: $60.

> **60**

---

**Ours (PRISM)**

If it costs Charlie $10.00 to make 10 candles, then:

**1. Calculate the cost of supplies for 20 candles:**
- Cost for 10 candles = $10.00
- Cost for 20 candles = $\$10.00 \times (20/10) = \$10.00 \times 2 = \$20.00$

**2. Calculate the selling price per candle:**
- Selling price per candle = $2.00

**3. Calculate the net profit per candle:**
- Net profit per candle = Selling price - Cost per candle
- Cost per candle = $\$10.00/10 = \$1.00$
- Net profit per candle = $\$2.00 - \$1.00 = \$1.00$

**4. Calculate the total net profit for 20 candles:**
- Total net profit = Net profit per candle $\times$ Number of candles
- Total net profit = $\$1.00 \times 20 = \$20.00$

The answer is: 20.

> **20**

### D.1.3. LLADA 2.0 MINI

---

**Query (GSM8K)**

Mishka bought 3 pairs of shorts, 3 pairs of pants, and 3 pairs of shoes. One pair of shorts costs $16.50. One pair of pants costs $22.50 and one pair of shoes costs $42. How many dollars did Mishka spend on all the clothing items?

---

**Original (LLaDA 2.0 mini)**

To find the total amount Mishka spent, we need to calculate the cost of each type of clothing separately and then add them up.

1. Calculate the cost of the shorts:
- Mishka bought 3 pairs of shorts. Each pair costs $16.50.
- Total cost = $3 \times 16.50 = 49.5$ dollars.

2. Calculate the cost of the pants:
- Mishka bought 3 pairs of pants. Each pair costs $22.5.
- Total cost = $3 \times 22.5 = 67.5$ dollars.

3. Calculate the cost of the shoes:
- Mishka bought 1 pair of shoes. Each pair costs $84.50.
- Total for shoes = $1 \times 84.50 = 84.50$ dollars.

By summing the expenses, the final expenditure on the primary items is determined to be 84.5 dollars.

**84.5**

---

**Ours (PRISM)**

Mishka bought 3 pairs of shorts, 3 pairs of pants, and 3 pairs of shoes.

- Cost of shorts = $3 \times \$16.50 = \$49.50$
- Cost of pants = $3 \times \$22.50 = \$67.50$
- Cost of shoes = $3 \times \$42.00 = \$126.00$

Total cost = (number of shorts $\times$ cost per pair) + (number of pants $\times$ cost per pair) + (number of shoes $\times$ cost per pair)
$= (3 \times \$16.50) + (3 \times \$22.50) + (3 \times \$42.00)$
$= \$49.50 + \$67.50 + \$126.00$
$= \$243.00$

Mishka spent 243 dollars on all the clothing items.

**243**

## D.2. HumanEval

### D.2.1. LLADA 8B INSTRUCT

---

**Query (HumanEval)**

```python
from typing import List

def string_xor(a: str, b: str) -> str:
    """ Input are two strings a and b consisting only of 1s and 0s.
    Perform binary XOR on these inputs and return result also as a string.
    >>> string_xor('010', '110')
    '100'
    """
```

---

**Original (LLaDA 8B Instruct)**

```python
def string_xor(a: str, b, str): str:
    result = []

    for i in range(min(len(a), max(b))):
        result.append(int(a[i]) ^ int(b[i]))

    return ''.join(result)
```

---

**Ours (PRISM)**

```python
def string_xor(a: str, b: str) -> str:
    result = []

    for char_a, char_b in zip(a, b):
        val = int(char_a) ^ int(char_b)
        result.append(str(val))

    return ''.join(result)
```

## D.2.2. DREAM 7B INSTRUCT

### Query (HumanEval)

```python
from typing import List

def mean_absolute_deviation(numbers: List[float]) -> float:
    """ For a given list of input numbers, calculate Mean Absolute Deviation
    around the mean of this dataset.
    Mean Absolute Deviation is the average absolute difference between each
    element and a centerpoint (mean in this case):
    MAD = average | x - x_mean |
    >>> mean_absolute_deviation([1.0, 2.0, 3.0, 4.0])
    1.0
    """
```

### Original (Dream 7B Instruct)

```python
def mean_absolute_deviation(numbers):
    mean_value = sum(numbers) / len(numbers)
    abs_differences = [abs(number - mean) for for number in numbers]
    mad_value = sum(abs_differences) / len(numbers)

    return mad_value
```

### Ours (PRISM)

```python
def mean_absolute_deviation(numbers):
    mean = sum(numbers) / len(numbers)
    abs_diff = [abs(number - mean) for number in numbers]
    mad = sum(abs_diff) / len(abs_diff)

    return mad
```

### D.2.3. LLADA 2.0 MINI

**Query (HumanEval)**

```python
def greatest_common_divisor(a: int, b: int) -> int:
    """ Return a greatest common divisor of two integers a and b
    >>> greatest_common_divisor(3, 5)
    1
    >>> greatest_common_divisor(25, 15)
    5
    """
```

**Original (LLaDA 2.0 mini)**

```python
def gcd(a, b):
    while b:
        a, b = b,        % a
    return a

# Example usage:
print(gcd(25, 15))
```

**Ours (PRISM)**

```python
def greatest_common_divisor(a, b):
    if b == 0:
        return a
    return greatest_common_divisor(b, a % b)
```

## D.3. Math500

### D.3.1. LLADA 8B INSTRUCT

---

**Query (MATH-500)**

Convert the point $(0, 3)$ in rectangular coordinates to polar coordinates. Enter your answer in the form $(r, \theta)$, where $r > 0$ and $0 \leq \theta < 2\pi$.

---

**Original (LLaDA 8B Instruct)**

To convert the point $(0, 3)$ from rectangular coordinates to polar coordinates, we need to use the following formulas:

$$r = \sqrt{x^2 + y^2}$$

$$\theta = \tan^{-1}\left(\frac{y}{x}\right)$$

Here, $x = 0$ and $y = 3$.

1. Calculate $r$:

$$r = \sqrt{0^2 + 3^2} = \sqrt{9} = 3$$

2. Calculate $\theta$:

$$\theta = \tan^{-1}\left(\frac{3}{0}\right) = \tan^{-1}(3)$$

**The value of $\tan^{-1}(3)$ is $\frac{\pi}{3}$ because $\tan\left(\frac{\pi}{3}\right) = 3$.**

Therefore, the polar coordinates of the point $(0, 3)$ are $(3, \frac{\pi}{3})$.

$$\boxed{(3, \tfrac{\pi}{3})}$$

---

**Ours (PRISM)**

To convert the point $(0, 3)$ from rectangular coordinates to polar coordinates, we need to find the radial distance $r$ and the angle $\theta$.

1. The radial distance $r$ is given by the distance formula:

$$r = \sqrt{x^2 + y^2}$$

For the point $(0, 3)$, $x = 0$ and $y = 3$, so:

$$r = \sqrt{0^2 + 3^2} = \sqrt{9} = 3$$

2. The angle $\theta$ is given by the arctangent function:

$$\theta = \tan^{-1}\left(\frac{y}{x}\right)$$

For the point $(0, 3)$, $x = 0$ and $y = 3$. Since the point lies on the positive $y$-axis, the angle is:

$$\theta = \frac{\pi}{2}$$

Therefore, the polar coordinates are $(3, \frac{\pi}{2})$.

$$\boxed{(3, \tfrac{\pi}{2})}$$

## D.3.2. DREAM 7B INSTRUCT

---

### Query (MATH-500)

What is the least positive integer multiple of 30 that can be written with only the digits 0 and 2?

---

### Original (Dream 7B Instruct)

**Reasoning:** To find the least positive integer multiple of 30 that can be written with only the digits 0 and 2, we need to find a number that is divisible by both 5 and 6, since 30 is the least common multiple of 5 and 6.

**Step 1:** First, let's find the smallest multiple of 5 that can be written with only the digits 0 and 2. This would be 20, since 20 is divisible by 5.

**Step 2:** Next, let's find the smallest multiple of 6 that can be written with only the digits 0 and 2. This would be 20 itself, since 20 is divisible by both 2 and 3.

**Step 3:** Now, we need to find a number that is divisible by both 20 and 20 (or equivalently, divisible by both 20 and 30). The smallest such number is 2020, which is divisible by 30.

Therefore, the least positive integer multiple of 30 that can be written with only the digits 0 and 2 is 2020.

**2020**

---

### Ours (PRISM)

The number must be divisible by $30 = 2 \cdot 3 \cdot 5$, so it must end in 0, divisible by 3, and divisible by 5.

We can start by listing the possible numbers that end in 0 and are divisible by 3. The possible numbers that end in 0 and are divisible by 3 are 2220, 22020, 20220, 22220, and 22020.

From these, we choose the smallest number that is divisible by 5. The possible numbers that end in 0 and are divisible by 3 are 2220, 22020, 20220, 22220, 22020, and 20220.

The smallest number that is divisible by 5 is 2220. Therefore, the least positive integer multiple of 30 that can be written with only the digits 0 and 2 is 2220.

**2220**

D.3.3. LLADA 2.0 MINI

---

### Query (MATH-500)

If the domain of the function $\log x^2$ is $x < a$ or $x > b$, for some $a$ and $b$, find $a + b$.

---

### Ours (PRISM)

The function is given by $f(x) = \log x^2$.

1. For the logarithm to be defined, its argument must be strictly positive:

$$x^2 > 0$$

2. This inequality is true for all real numbers $x$ except $x = 0$. Therefore, the domain of the function is $x \neq 0$.

3. We can write this domain using inequalities as $x < 0$ or $x > 0$.

4. The problem states that the domain is $x < a$ or $x > b$. By comparing this with our derived domain, we can identify:

$$a = 0 \quad \text{and} \quad b = 0$$

5. The question asks for the value of $a + b$:

$$a + b = 0 + 0 = 0$$

$$\boxed{0}$$

---

### Original (Dream 7B Instruct)

We are given the function $\log x^2$.

**Step 1:** The function is $\log x^2$. Note that $x^2$ is always non-negative for real $x$, and it is zero only when $x = 0$.

**Step 2:** The logarithm of a real number is defined only when the argument is positive. So we require:

$$x^2 > 0$$

**Step 3:** This inequality holds for all real $x$ except $x = 0$, since $0^2 = 0$, and $\log(0)$ is undefined.

**Step 4:** So the domain is all real numbers $x$ such that:

$$x \neq 0$$

$$\boxed{(\text{-}\infty, 0) \cup (0, \infty)}$$

**D.4. MBPP**

D.4.1. LLaDA 8B Instruct

> **Query (MBPP)**
>
> Write a function to find whether a given array of integers contains any duplicate element.
> Your code should pass the test:
>
> ```python
> assert test_duplicate([1, 2, 3, 4, 5]) == False
> ```

**Original (LLaDA 8B Instruct)**

```python
def test_duplicate(arr):
    seen = set()

    for num in arr:
        seen.add(num)

    if num in seen:
        return True
    return False
```

**Ours (PRISM)**

```python
def test_duplicate(arr):
    seen = set()

    for num in arr:
        if num in seen:
            return True
        seen.add(num)

    return False
```

### D.4.2. DREAM 7B INSTRUCT

---

**Query (MBPP)**

Write a function to find the largest palindromic number in the given array.
Your code should pass the test:

```
assert largest_palindrome([1, 232, 54545, 999991], 4) == 54545
```

---

**Original (Dream 7B Instruct)**

```python
def largest_palindrome(arr, n):
    for i in range(n):
        if str(arr[i]) == str(arr[i])[::-1]:
            return arr[i]
    return -1
```

**Ours (PRISM)**

```python
def largest_palindrome(arr, n):
    largest_palindrome = -1

    for i in range(n):
        if str(arr[i]) == str(arr[i])[::-1] and arr[i] > largest_palindrome:
            largest_palindrome = arr[i]

    return largest_palindrome
```

---

### D.4.3. LLADA 2.0 MINI

---

**Query (MBPP)**

Write a function to find whether a given array of integers contains any duplicate element.
Your code should pass the test:

```
assert test_duplicate([1, 1, 2, 2, 3]) == True
```

---

**Original (LLaDA 2.0 mini)**

```python
def contains_duplicate(arr):
    seen = set()
    for element in arr:
        if element in seen:
            return True
        seen.add(element)
    return False
```

**Ours (PRISM)**

```python
def test_duplicate(arr):
    return len(arr) != len(set(arr))
```

