# OpenReview forum: "Efficient Test-Time Scaling via Hierarchical Search and Self-Verification for Discrete Diffusion Language Models"
_ICML.cc/2026/Conference — ICML 2026 regular_

### Official Review · Reviewer_m9jQ · 2026-03-11

**Soundness:** 2
**Presentation:** 3
**Significance:** 3
**Originality:** 3
**Overall Recommendation:** 5
**Confidence:** 4

**Summary:**

This paper proposes LLaDA-S, an efficient test-time scaling (TTS) framework specifically designed for discrete diffusion language models (dLLMs). The work makes three inter-related contributions: Hierarchical Trajectory Search (HTS), which dynamically prunes the active trajectory pool via a geometric decay schedule concentrated in the early-to-mid denoising window; Self-Verified Feedback (SVF), which repurposes the same dLLM as a lightweight binary verifier by scoring intermediate completions through a Yes/No prompt; and local branching via partial remasking, which diversifies the surviving trajectory pool by selectively re-masking low-confidence token positions while preserving a high-confidence logic skeleton. Evaluated across four benchmarks (GSM8K, MATH500, HumanEval, MBPP) and three dLLM families, LLaDA-S achieves NFE reductions of 1.7×–6.5× relative to Best-of-N at comparable accuracy levels.

**Compliance With Llm Reviewing Policy:**

Affirmed.

**Final Justification:**

The rebuttal addressed most of my main concerns. Consequently, I raise my score to accept from weak accept.

**Key Questions For Authors:**

Q1. Can you report the precision, recall, and F1 of SVF Yes/No judgments against ground truth correctness labels, measured at representative timesteps within the pruning window on GSM8K and MATH500? Additionally, what fraction of pruning decisions at K=8 on MATH500 discard the trajectory that would have produced the correct final answer?

Q2. Can you provide a direct performance and efficiency comparison between SVF and existing dLLM-compatible reward-guided methods, including RFG [1] and  Reward-Guided Stitching [2] ? Specifically, what accuracy does LLaDA-S achieve when SVF is replaced by these alternatives under matched NFE budgets, and how does total wall-clock time compare? This comparison is necessary to establish the practical and qualitative boundaries of SVF's contribution relative to available alternatives.

Q3.  The entropy curves in Appendix A appear to show continuous, gradual degradation rather than a sharp early drop followed by a stable mid-window plateau. How do the authors reconcile this observation with the claim that entropy dynamics specifically justify W = [0.1, 0.6]? Was this window selected by the entropy analysis or by grid search?

I appreciate the paper's core ideas and the quality of its experimental design, balanced against the unresolved questions about SVF reliability. A targeted rebuttal addressing Q1–Q3 would provide sufficient basis to raise this score to a clear accept. Conversely, evidence that SVF does not reliably discriminate at intermediate timesteps and that gains derive primarily from HTS alone would warrant reassessment of the contribution's novelty.

[1] Chen, Tianlang, et al. "Rfg: Test-time scaling for diffusion large language model reasoning with reward-free guidance." arXiv preprint arXiv:2509.25604 (2025).
[2] Miles, Roy, et al. "Test-Time Scaling with Diffusion Language Models via Reward-Guided Stitching." arXiv preprint arXiv:2602.22871 (2026).

**Limitations:**

Resolving the Q1~Q3 would additionally verify the contribution of the proposed work.

**Strengths And Weaknesses:**

**S1. Clear Motivation and Reasonable Structure.**
The author's indication of limitations of current dLLM works and suggestion of novel solution of handling those problem with Hierarchical Trajectory Search (HTS),  Self-Verified Feedback (SVF), and local branching shows the novelty and clear contribution of the suggested method.

**S2. Elimination of external verifiers has genuine practical value.**
The SVF design — using the generator model itself for binary judgment — avoids loading a separate reward model, which meaningfully reduces deployment memory. Table 3 shows SVF is competitive with Qwen2-7B-level external verifiers while keeping the total parameter footprint at 8B. This is a real and reproducible advantage, not merely a motivational claim.

**S3. Hyperparameter ablation is thorough and honest.**
The systematic sweep across all five HTS hyperparameters (W, d, i, S, K) in Appendix B, replicated across four benchmarks, is commendable. The results clearly identify the pruning window and decay factor as the most sensitive choices, and the reported defaults do not appear cherry-picked from outlier configurations.

**W1. SVF reliability as a verifier is entirely uncharacterized**
The entire LLaDA-S pipeline depends on SVF producing a reliable ranking signal during progressive thinning. Yet the paper provides no direct evidence that SVF binary judgments correlate with solution correctness at intermediate denoising steps. Table 1 reports downstream task accuracy, and Table 3 compares against external verifiers on end-task performance — neither measures SVF's discriminative quality directly. More critically, dLLMs operating at base accuracy of only 26.40% on MATH500 are frequently and confidently wrong. In this regime, a self-verifier exhibiting self-consistency bias — assigning high scores to its own systematically wrong outputs — would cause HTS to prune correct trajectories preferentially. This failure mode is not analyzed anywhere in the paper.

**W2. The absence of direct comparison between SVF and established external reward models.** The absence of direct comparison between SVF and external reward model obscures the true contribution of the proposed verifier.
While avoiding external reward models is a stated motivation for SVF, a principled evaluation of what is gained and lost relative to existing dLLM-specific reward-guided methods is missing.

**W3. The entropy analysis does not convincingly justify the pruning window placement.**
The paper repeatedly cites Appendix A as empirical motivation for the mid-denoising pruning window W = [0.1, 0.6]. However, inspecting Figures 4–7 reveals that the entropy curves show a largely continuous, monotonic degradation from high to low entropy across all timesteps — not the two-phase structure ("sharp early drop followed by a stable plateau of competing trajectories") the authors describe. If entropy decreases gradually and uniformly, it provides no principled argument for concentrating pruning specifically in the [0.1, 0.6] window rather than anywhere else. The window choice thus appears to be empirically tuned rather than theoretically motivated, which undermines the paper's design narrative.

---

> ### Author Rebuttal · Authors · 2026-03-31
>
> We sincerely thank the reviewer for the constructive feedback. We address each point below.
>
> **W1 & Q1: SVF reliability and self-consistency bias.**
>
> We believe the concern rests on a misunderstanding of what SVF evaluates. SVF does not judge whether the final answer is correct against ground truth. It is applied to intermediate completions $\hat{z}_0$ during denoising, asking "Is this solution path sound and likely to result in the correct final answer?" (Appendix C), assessing reasoning coherence, not answer correctness. SVF is used for relative ranking among competing trajectories at each pruning step, not as an absolute correctness oracle: it only needs to distinguish more coherent paths from less coherent ones.
>
> Since SVF operates on intermediate states with no final answer, we cannot measure accuracy against dataset ground truth. We instead use Qwen3-235B-A22B-Instruct as a much stronger proxy evaluator:
>
> | Benchmark | Eval Point | F1 ($K$=2/4/8) | ECE ($K$=2/4/8) |
> |-----------|------|---:|---:|
> | GSM8K | First SVF | .676 / .696 / .731 | .330 / .281 / .231 |
> | GSM8K | Last SVF | .649 / .631 / .639 | .286 / .170 / .091 |
> | MATH-500 | First SVF | .650 / .697 / .704 | .306 / .267 / .204 |
> | MATH-500 | Last SVF | .634 / .629 / .631 | .258 / .158 / .097 |
>
> Because the pruning stage retains a fixed K, the number of positives is fixed, which makes Precision, Recall, and F1 numerically identical. P/R/F1 of 0.63–0.73 is well above random. Critically, SVF achieves comparable P/R/F1 on MATH-500 (0.63–0.70) and GSM8K (0.63–0.73) despite a 2.5× gap in base accuracy (26.4% vs. 67.6%). If self-consistency bias dominated, SVF would collapse on harder tasks where the model is more frequently wrong, but this is not what we observe.
>
> Regarding the percentage of pruning decisions that discard correct trajectories: since SVF operates on intermediate denoising states, each pruned trajectory does not uniquely correspond to a single final output but rather represents a distribution over many possible continuations. It is therefore not well-defined to label an intermediate trajectory as correct or incorrect without committing to a specific continuation. However, we can provide strong indirect evidence: LLaDA-S achieves 42.80% on MATH-500, surpassing Best-of-16 (38.00%) with ~3× fewer NFE. If SVF systematically discarded correct trajectories, it could not outperform Best-of-$N$, which retains all trajectories. Additionally, replacing SVF with random pruning on MATH-500 consistently degrades performance (up to 4.7% at $K=8$, see ablation in our response to Reviewer 5cBX), confirming SVF provides a meaningful ranking signal.
>
> **W2 & Q2: Comparison with external reward-guided methods.**
>
> We replace SVF with RFG and Reward-Guided Stitching within the LLaDA-S framework on GSM8K (LLaDA-8B):
>
> | $K$ | Verifier | Pass@1 (%) | NFE | Time (s) | Params |
> |-----|----------|------:|----:|------:|---:|
> | 2 / 4 / 8 | SVF | 74.24 / 75.30 / 85.30 | 283 / 509 / 1048 | 5.40 / 8.92 / 17.08 | 8B |
> | 2 / 4 / 8 | RFG | 70.68 / 71.59 / 81.29 | 291 / 519 / 1073 | 9.08 / 15.70 / 32.27 | 16B |
> | 2 / 4 / 8 | Stitching | 77.12 / 81.59 / 90.68 | 319 / 570 / 1158 | 5.68 / 8.90 / 17.07 | 15B |
>
> We note that this comparison is not entirely fair: both alternatives rely on additional oracle information that SVF does not require. RFG computes its verifier score from the log-likelihood ratio between a policy model and a reference model, requiring both models loaded simultaneously (16B params total). Stitching uses an external process reward model (PRM) to score intermediate reasoning steps (15B params total). In contrast, SVF operates using only the policy model with no task-specific training or external models.
>
> Despite this asymmetry, SVF consistently outperforms RFG in accuracy, wall-clock time, and memory. Stitching achieves higher accuracy but at nearly 2× the parameter footprint and with reliance on a task-specific PRM, limiting its generality. We view these as complementary points in the deployment–accuracy trade-off space. We also note that Reward-guided Stitching (26 Feb 2026) is a concurrent work of our submission.
>
> **W3 & Q3: Entropy analysis and pruning window.**
>
> We would like to clarify that we do not claim the entropy curves exhibit a sharp phase transition that uniquely determines $W=[0.1, 0.6]$. The entropy analysis provides qualitative intuition: very early steps remain too noisy for reliable pruning, while very late steps are dominated by low-entropy token refinement where pruning brings less benefit. The entropy trends suggest pruning should happen in an intermediate window, but do not specify a unique optimum. Our final choice of $W=[0.1, 0.6]$ was selected empirically, and the entropy plots are intended as supporting evidence that this choice is reasonable, rather than as a formal justification. We will revise the paper to make this distinction explicit.

---

> > ### Author Rebuttal · Reviewer_m9jQ · 2026-04-01
> >
> > All of my concerns seems to be addressed. I will update my score to accept.

---

> > > ### Author Response · Authors · 2026-04-06
> > >
> > > We are glad that our rebuttal has adequately addressed the reviewer's concerns. We sincerely thank the reviewer for raising the score and for the constructive engagement throughout the discussion. The discussion and feedback has been very helpful and has greatly strengthened the paper.

---

### Official Review · Reviewer_5cBX · 2026-03-13

**Soundness:** 3
**Presentation:** 3
**Significance:** 3
**Originality:** 3
**Overall Recommendation:** 4
**Confidence:** 4

**Summary:**

The authors tackle the problem of test time compute scaling for discrete diffusion language models (dLLMs). Since traditional autoregressive scaling relies on left to right search trees, it is fundamentally incompatible with the parallel and global denoising process of dLLMs. To bridge this gap, the authors introduce LLaDA-S. The framework uses Hierarchical Trajectory Search to dynamically allocate compute during the critical early to mid denoising window. To avoid the massive memory overhead of external reward models, they propose Self Verified Feedback, which prompts the base dLLM to judge its own intermediate hypotheses. Finally, they use local branching with partial remasking to explore diverse reasoning paths while preserving high confidence tokens. They evaluate on reasoning and code benchmarks like GSM8K, MATH500, and HumanEval, showing it matches Best of N performance at a fraction of the computational cost (NFE).

**Compliance With Llm Reviewing Policy:**

Affirmed.

**Key Questions For Authors:**

1. How does the Self Verified Feedback mechanism avoid the trap of inherent model overconfidence, specially on complex MATH500 problems where the base model lacks the capacity to verify its own logic?

2. Can you provide a detailed breakdown of the end to end wall clock latency for LLaDA-S compared to the autoregressive baselines, factoring in the sequential overhead of the verification prompts?

**Limitations:**

The authors need to address the vulnerability of their system to self confirmation bias during the verification stage.

**Strengths And Weaknesses:**

Strengths:

- The parallel generation dynamics of dLLMs theoretically offer better global planning capabilities, and this paper provides a good step to harness that.

- The Hierarchical Trajectory Search mechanism is intuitive. Concentrating compute on the early to mid denoising window makes sense. Late stage denoising is mostly deterministic token fixing rather than high level semantic reasoning, so pruning early saves useless compute.

- The memory efficiency gained by using Self Verified Feedback instead of a seperate 7B or 8B reward model is a massive practical win for deployment on standard hardware.

Weaknesses:

- The reliance on Self Verified Feedback is heavily problematic for harder tasks. While prompting a model to self evaluate works okay on GSM8K, LLMs are notoriously bad at catching their own subtle logical flaws in harder domains like MATH500. A model that hallucinates a complex math proof will likely hallucinate a "Yes" judgement for it.

- The paper claims to preserve high confidence tokens during partial remasking, but the heuristic used to determine "high confidence" during intermediate diffusion steps is quite brittle. Diffusion entropy curves vary wildly across different prompt complexities. This means a static confidence threshold will inevitably mask out correct reasoning steps on hard problems.

---

> ### Author Rebuttal · Authors · 2026-03-31
>
> We sincerely thank the reviewer for the constructive feedback. We address each point below.
>
> **W1 & Q1: SVF reliability on harder tasks and self-confirmation bias.**
>
> We believe the concern rests on a misunderstanding of what SVF evaluates. SVF does not ask the model whether its final answer is correct. It is applied to intermediate completions $\hat{z}_0$ during denoising, asking "Is this solution path sound and likely to result in the correct final answer?" (Appendix C), assessing reasoning coherence, not answer correctness. A hallucinated proof with logical gaps will score lower than a coherent one, even if both ultimately lead to wrong answers. Crucially, SVF is used for relative ranking among competing trajectories at each pruning step, not as an absolute correctness oracle: it only needs to distinguish better from worse paths, not certify correctness.
>
> We verify this empirically. First, replacing SVF with random pruning on MATH-500 consistently degrades performance:
>
> | Config | $K=2$ | $K=4$ | $K=8$ |
> |--------|------:|------:|------:|
> | SVF pruning | 30.16 | 37.70 | 42.80 |
> | Random pruning | 29.76 ($\Delta$-0.40) | 35.12 ($\Delta$-2.58) | 38.10 ($\Delta$-4.70) |
>
> Second, we measure SVF's agreement with a much stronger proxy evaluator (Qwen3-235B-A22B-Instruct) at the first and last SVF evaluations within the pruning window:
>
> | Benchmark | Eval Point | F1 ($K$=2/4/8) | ECE ($K$=2/4/8) |
> |-----------|------|---:|---:|
> | GSM8K | First SVF | .676 / .696 / .731 | .330 / .281 / .231 |
> | GSM8K | Last SVF | .649 / .631 / .639 | .286 / .170 / .091 |
> | MATH-500 | First SVF | .650 / .697 / .704 | .306 / .267 / .204 |
> | MATH-500 | Last SVF | .634 / .629 / .631 | .258 / .158 / .097 |
>
> ECE for Expected Calibration Error. SVF achieves comparable F1 on MATH-500 (0.63–0.70) and GSM8K (0.63–0.73), despite a 2.5× gap in base accuracy (26.4% vs. 67.6%). If self-confirmation bias dominated, we would expect SVF to collapse on harder tasks where the model is more frequently wrong, but this is not what we observe. ECE also consistently improves from first to last SVF evaluation, indicating that SVF confidence becomes better calibrated as denoising progresses.
>
> **W2: Partial remasking confidence threshold.**
>
> We would like to clarify that the "high-confidence" tokens in partial remasking are not determined by a fixed confidence threshold. At each step, we compute token-wise confidence scores and form a relative low-confidence pool based on ranking. We then randomly remask only a subset from this pool, preserving the rest as the current logic skeleton. This relative design does not rely on a static global threshold and is therefore less sensitive to variations in entropy scale across prompts and denoising stages. Additionally, we only remask a random subset of the low-confidence pool (Algorithm 2: $I_t \subseteq U_t$), not the entire pool, which further reduces the risk of accidentally masking out correct reasoning tokens.
>
> **Q2: Wall-clock time.**
>
> We provide a wall-clock breakdown on GSM8K (LLaDA-8B), factoring in SVF overhead:
>
> | $K$ | Pass@1 | NFE | SVF Calls | Inf. (s) | SVF (s) | Total (s) |
> |-----|------:|----:|----------:|------:|------:|------:|
> | 2 | 74.24 | 283 | 27 | 4.63 | 0.54 | 5.17 |
> | 4 | 75.30 | 509 | 29 | 7.95 | 0.58 | 8.53 |
> | 8 | 85.30 | 1048 | 33 | 15.79 | 0.67 | 16.46 |
>
> A single SVF call costs 0.020s vs. 0.016s for a denoising step. SVF is applied sparsely (27–33 calls), so its overhead is only 4–10% of total runtime. We do not include AR baselines as the goal of this work is efficient test-time scaling within the dLLM paradigm; a fair dLLM-vs-AR wall-clock comparison would require controlling for many additional factors (e.g., KV-cache) and is beyond our current scope.

---

> > ### Author Rebuttal · Reviewer_5cBX · 2026-04-04
> >
> > Thanks for your rebuttal, but I increase my score to 5, accept

---

> > > ### Author Response · Authors · 2026-04-06
> > >
> > > We are glad that our rebuttal has adequately addressed the reviewer's concerns. We sincerely thank the reviewer for the constructive engagement throughout the discussion. The discussion and feedback has been very helpful and has greatly strengthened the paper.
> > >
> > > We would greatly appreciate it if the reviewer could update the score to accept to reflect the resolution of the discussion.

---

### Official Review · Reviewer_zQbn · 2026-03-13

**Soundness:** 2
**Presentation:** 3
**Significance:** 2
**Originality:** 2
**Overall Recommendation:** 4
**Confidence:** 4

**Summary:**

LLaDA-S is a test-time scaling framework for discrete diffusion LMs. It prunes parallel denoising trajectories via a geometric decay schedule (HTS), scores them using the same dLLM as a binary Yes/No verifier (SVF), and branches survivors by remasking low-confidence tokens. Evaluated on math/code benchmarks across three dLLMs, it matches best-of-N at fewer function evaluations.

**Compliance With Llm Reviewing Policy:**

Affirmed.

**Final Justification:**

Authors' response addressed part of my concerns and also shed some interesting differences for ITS in autogressive LMs vs diffusion LMs therefore I raised my score to 4.

**Key Questions For Authors:**

* How does LLaDA-S relate to PG-DLM? Can you provide a direct empirical comparison on shared benchmarks (e.g., LLaDA-8B)? If LLaDA-S reaches a comparable Pareto frontier, I'm happy to raise my score.
* What is SVF's actual verification accuracy against ground truth, especially in low-accuracy regimes like MATH-500?
* Does partial remasking preserve or distort the model's learned distribution? Any distributional guarantees?

**Limitations:**

yes

**Strengths And Weaknesses:**

Strengths

* Timely problem — TTS for dLLMs is underexplored and the area is moving fast.
* SVF avoids loading a separate verifier, which is a real practical win for deployment.
* Solid experimental coverage: three models, four benchmarks, thorough hyperparameter sweeps.
* The entropy analysis nicely motivates where pruning should happen in the denoising trajectory.

Weaknesses

* Novelty vs. PG-DLM (Dang et al., 2025; arXiv:2507.08390). This concurrent work formalizes the same problem using particle Gibbs with conditional SMC, with convergence guarantees and principled compute allocation analysis. HTS is essentially an ad-hoc particle filter with geometric thinning; partial remasking resembles conditional resampling around a reference trajectory. The paper doesn't cite or discuss this connection at all.
* No theory. All design choices (decay schedule, window, branching) are heuristic. The extensive appendix sweeps are doing the work that formal analysis should provide.
* SVF reliability is unexamined. No calibration analysis or accuracy breakdown. On MATH-500 where the base model gets ~26%, how reliable is a weak model's self-assessment?
* Evaluation limited to verifiable tasks (math/code). Unclear how SVF generalizes to open-ended generation.
* Wall-clock time mostly missing — NFE doesn't account for the longer prompts in SVF calls.

---

> ### Author Rebuttal · Authors · 2026-03-31
>
> We sincerely thank the reviewer for the constructive feedback. We address each point below.
>
> **W1 & Q1: Relationship to PG-DLM**
>
> We apologize for the missing citation and will add a discussion in the revision. We respectfully argue that HTS is not an "ad-hoc particle filter"; the two differ at the formulation level.
>
> PG-DLM is distributional inference: it targets a reward-tilted distribution via importance weighting at every denoising step, with convergence guarantees. LLaDA-S is combinatorial search: find the best trajectory under a specific NFE budget. SVF scores are heuristic ranking signals, not importance weights from any density ratio, and there is no $p^*$ that HTS converges to. Particle filtering requires (i) density-ratio weights at every step and (ii) resampling by duplicating particles. HTS satisfies neither: partial remasking modifies trajectories via local mutation, not duplication. By this criterion, beam search and MCTS would equally be "particle filters".
>
> We ran LLaDA-S on PG-DLM's CoLA benchmark with the same external classifier:
>
> | Method | $k$ or $K$ | Acc. (%) | NFE |
> |--------|-----------|------:|------:|
> | PG-DLM | 4 / 16 / 64 | 77.8 / 91.1 / 90.6 | **200** / 800 / 3200 |
> | LLaDA-S | 2 / 4 / 8 | **84.3** / **92.3** / **93.7** | 324 / **662** / **1024** |
>
> LLaDA-S ($K=4$) surpasses PG-DLM ($k=16$) in both accuracy and NFE, while PG-DLM *decreases* at $k=64$, confirming that LLaDA-S achieves a superior Pareto frontier on this benchmark.
>
> **W2: No theory**
>
> LLaDA-S is a search procedure, not distributional inference (W1). The criterion for search is *empirical robustness*, not convergence guarantees, the same standard applied to beam search, MCTS, and nucleus sampling. A single set of defaults ($W{=}[0.1, 0.6]$, $d{=}1.8$, $i{=}3$) transfers across all 4 benchmarks and 3 models without task-specific tuning, which we believe demonstrates sufficient robustness for a search-based method.
>
> **W3 & Q2: SVF reliability**
>
> We believe there is a misunderstanding of SVF. It does not verify whether the final answer matches ground truth. SVF is applied to intermediate completions during denoising, asking "Is this solution path sound and likely to result in the correct final answer? (Appendix C) — assessing reasoning coherence, not answer correctness, and used for relative ranking among trajectories rather than as an absolute oracle. Since intermediate states have no final answer to compare against dataset ground truth, we use a much stronger model (Qwen3-235B-A22B-Instruct) as proxy evaluator:
>
> | Benchmark | $K$ | Eval Point | F1 | ECE |
> |-----------|-----|-----:|-----:|-----:|
> | GSM8K | 2 / 4 / 8 | First SVF | 0.676 / 0.696 / 0.731 | 0.330 / 0.281 / 0.231 |
> | GSM8K | 2 / 4 / 8 | Last SVF | 0.649 / 0.631 / 0.639 | 0.286 / 0.170 / 0.091 |
> | MATH-500 | 2 / 4 / 8 | First SVF | 0.650 / 0.697 / 0.704 | 0.306 / 0.267 / 0.204 |
> | MATH-500 | 2 / 4 / 8 | Last SVF | 0.634 / 0.629 / 0.631 | 0.258 / 0.158 / 0.097 |
>
> ECE for Expected Calibration Error. Critically, SVF achieves comparable F1 on MATH-500 (0.63–0.70) and GSM8K (0.63–0.73), despite a 2.5× gap in base accuracy (26.4% vs. 67.6%), showing that its discriminative ability is not bottlenecked by the model's task-solving capacity. Calibration also improves from first to last SVF evaluation (lower ECE), as intermediate completions become more informative.
>
> **Q3: Partial remasking**
>
> As discussed in W1, LLaDA-S is a search procedure whose goal is to find the best trajectory, not to preserve $p_\theta$. Partial remasking diversifies search by perturbing low-confidence tokens while retaining a high-confidence skeleton. Remasked tokens are denoised by the same $p_\theta$ from $\tilde{z}_t = \text{Remask}(z_t; I_t)$, so generation remains within the model's learned conditional. The criterion is downstream performance, not distributional fidelity, and remasking consistently improves results across all benchmarks (see ablation in our response to Reviewer UBqU), suggesting it is an effective exploration mechanism that does not introduce harmful artifacts.
>
> **W4: Open-ended generalization**
>
> Since SVF evaluates reasoning coherence rather than ground-truth correctness (W3), it does not inherently require verifiable answers. We have evidence beyond math/code: LLaDA-S with SVF improves over baseline and ReMDM on TruthfulQA (Table 2), an open-ended factuality benchmark. We agree that broader evaluation on open-ended tasks is a valuable future direction.
>
> **W5: Wall-clock time**
>
> We provide a wall-clock breakdown on GSM8K (LLaDA-8B):
>
> | $K$ | Pass@1 | NFE | SVF Calls | Inf. (s) | SVF (s) | Total (s) |
> |-----|-------:|----:|----------:|--------------:|-------------:|----------:|
> | 2 | 74.24 | 283 | 27 | 4.63 | 0.54 | 5.17 |
> | 4 | 75.30 | 509 | 29 | 7.95 | 0.58 | 8.53 |
> | 8 | 85.30 | 1048 | 33 | 15.79 | 0.67 | 16.46 |
>
> A single SVF call costs 0.020s vs. 0.016s for a denoising step. SVF is applied sparsely (27–33 calls), so its overhead is only 4–10% of total runtime.

---

> > ### Author Rebuttal · Reviewer_zQbn · 2026-04-06
> >
> > Thanks for the detailed rebuttal. A few thoughts:
> > The CoLA comparison is appreciated and does address my main ask --- LLaDA-S looks competitive on that benchmark. That said, a single-benchmark comparison is limited. It would be more convincing repeated on math/code tasks where both methods have existing infrastructure and results. PG-DLM's drop at m=64 could also be a tuning issue on their end rather than a fundamental limitation.
> > I accept the "search vs. inference" distinction at the technical level --- HTS doesn't use importance weights, so calling it a particle filter is imprecise on my part. But the structural similarities (population → score → select → diversify) are real, and the paper should discuss the relationship to SMC-style methods in the revision regardless of framing. The missing citation alone warrants this.
> > The SVF calibration data is genuinely useful and should be in the main paper. F1 of ~0.65–0.70 with improving ECE over time is a reasonable signal for relative ranking. This substantially addresses my concern about low-accuracy regimes.
> > Wall-clock breakdown resolves that concern --- SVF overhead is clearly negligible.
> > I'm raising my score given the CoLA result and SVF calibration data, though I'd still like to see (a) explicit positioning vs. PG-DLM and SMC-style methods in the revision, and (b) the calibration analysis promoted from rebuttal to paper.
> >
> > One more remark: recent work on AR models (Puri et al., 2025, "Rollout Roulette") shows that probabilistic inference via particle-based MC methods can match or outperform search-based TTS for autoregressive LLMs --- diversity through sampling beats structured search. Your results seem to point in the opposite direction for dLLMs: search-style pruning and guided selection outperforms naive width scaling. This is an interesting contrast worth discussing. Intuitively, AR decoding is locally exact at each step so the bottleneck is mode coverage (sampling helps), whereas dLLMs are inherently stochastic with high early-step entropy (your Appendix A), so the bottleneck is concentrating compute on promising trajectories (search helps). You already have the entropy evidence for this argument but never make the conceptual case for *why* search is the right TTS paradigm for dLLMs when the AR literature suggests the opposite. Articulating this distinction would strengthen the paper's positioning considerably.

---

> > > ### Author Response · Authors · 2026-04-06
> > >
> > > We thank the reviewer for the continued engagement in the discussion. We are glad that our SVF calibration analysis and additional experiments on CoLA helped resolve the reviewer's concerns. We really appreciate that the reviewer is willing to raise the score given the CoLA result and SVF calibration data.
> > >
> > > We will include a dedicated section to discuss PG-DLM and related SMC-based inference methods for dLLMs and their technical connections and distinctions with our proposed LLaDA-S, for a better positioning of the method in the literature. We will also promote the calibration analysis of SVF into the main paper in the updated manuscript. Apart from these, we will conduct more experiments on math and coding tasks for a more comprehensive comparison between search-based and inference-based methods.
> > >
> > > Regarding the question of why search-based methods outperform probabilistic inference on dLLMs while the conclusion is the opposite for AR models, we think that the nature of diffusion LLMs as random-order autoregressive models (ROAR) plays a major role here. As pointed out in [1], diffusion LLMs naturally learn multiple possibilities for decoding trajectories during pretraining, yet this advantage remains buried unless we discover it at inference time. In other words, diffusion LLMs have a separate design space of generation order apart from guided generation with twisted transition probabilities. We believe that search-based methods for dLLMs effectively unlock this potential by finding better decoding trajectories that are already learned during pretraining. That said, we also genuinely believe that SMC-based probabilistic inference and search-based methods are not mutually exclusive but rather orthogonal directions in the design space for dLLMs. Developing a new method that takes the best of both worlds would be an exciting direction for future work. We will include these discussions in the paper for a better motivation of the method.
> > >
> > > [1] Kim, Jaeyeon, et al. "Train for the worst, plan for the best: Understanding token ordering in masked diffusions." ICML 2025

---

### Official Review · Reviewer_UBqU · 2026-03-20

**Soundness:** 2
**Presentation:** 3
**Significance:** 3
**Originality:** 2
**Overall Recommendation:** 5
**Confidence:** 4

**Summary:**

This paper proposes a test-time scaling framework for discrete masked diffusion models. It relies on a combination of self-verification with the dLLM as its own binary verifier, hierarchical search, and remasking. The authors show that their proposed method matches best-of-N generation with much fewer NFE across multiple datasets and models.

**Compliance With Llm Reviewing Policy:**

Affirmed.

**Final Justification:**

The rebuttal addressed the main concerns. The component ablation confirms SVF is the primary contributor (3.5-4.3pp over random pruning), with remasking adding a smaller but consistent gain. The SVF calibration analysis shows F1 of 0.63-0.73 with ECE improving over denoising steps, which is sufficient to confirm SVF provides a useful ranking signal beyond random pruning. The clarifications on selection mechanism, argmax vs. sampling, and the S=2 typo resolve the remaining ambiguities. The core limitations remain: the method is heuristic without formal guarantees, hyperparameter sensitivity can be steep in certain regimes, and evaluation is limited to verifiable tasks. That said, the paper addresses a timely problem with a practical solution that is well-validated across multiple models and benchmarks. Raising score from 4 to 5.

**Key Questions For Authors:**

1. Could you provide a minimal component ablation on at least one benchmark, particularly HTS with random pruning vs. SVF-guided pruning.

2. Could you report the SVF accuracy or ranking quality over timesteps? How does it compare to the final SVF on the completed sequence? I.e. could you provide some insights into how accurate the $z_0$ estimate is for the final outcome.

3. Could you clarify the final selection mechanism used for each benchmark and each method? Is there a setting where you execute code for LLaDa-S while this is not the case for best-of-N?

4. What is the selection criteria for best-of-N on each of the tasks? Did you try using the SVF reward as the selection criteria? This would be an important ablation to include in the paper.

5. For the SVF do you use argmax or sampling? What performs better?

6. How should practitioners set the pruning window and decay hyperparameters for a new task without a validation set?

**Limitations:**

yes

**Strengths And Weaknesses:**

Strengths:

- The paper addresses a timely and underexplored problem. The authors clearly articulate why AR-based TTS methods do not transfer to the parallel denoising paradigm, and propose a new efficient test-time scaling method.

- The entropy analysis (Appendix A) provides principled motivation for the pruning window placement rather than leaving it as an arbitrary design choice.

- The hyperparameter sweeps across all four benchmarks (Figures 8-11) provide a transparent view of the framework's robustness.


Weaknesses:

- The paper proposes three mechanisms but never isolates their contributions. Without ablations (e.g., HTS with random pruning, HTS+SVF without branching), it is not possible to determine which components drive the efficiency gains. It would great to include ablations to disentangle the contributions.

- The authors do not analyze the SVF, e.g. through classification metrics (precision, recall, calibration). Additionally, it would be even more interesting to see how SVF evolves over a typical denoising trajectory for each step. It largely relies on the choice of the $z_0$ prediction as a proxy for the outcome of the trajectory. This is a bit underdiscussed in the manuscript, how accurate is this estimate? How does the reward over denoising steps relate to the actual final reward? These are all important considerations that could be discussed in more detail.

- Large Hyperparameter sensitivity. On the positive side, a single set of default hyperparameters (W=0.1–0.6, d=1.8, i=3) appears to transfer across all four benchmarks and three models, which suggests reasonable robustness. However, the performance landscape around these defaults can be steep: on MBPP, shifting the pruning window from [0.1, 0.6] to [0.2, 0.7] causes a ~9-point Pass@1 drop (Table 19). Additionally, the main results in Table 1 use S=2 while the appendix analysis defaults to S=4, making it unclear which configuration is recommended in practice.

- The final selection mechanism is ambiguous for code benchmarks. If LLaDA-S uses code execution while the baseline uses majority voting, the comparison is not fair.

Minor: Appendix Figure 8 wrong caption

---

> ### Author Rebuttal · Authors · 2026-03-31
>
> We sincerely thank the reviewer for the constructive feedback. We address each point below.
>
> **W1 & Q1: Component ablation.**
>
> We ablate SVF and partial remasking independently on GSM8K (LLaDA-8B), keeping the same $K$ and NFE:
>
> | |K=2|K=4|K=8|
> |--------|------:|------:|------:|
> | Full (SVF + Remask) | 74.24 | 75.30 | 85.30 |
> | Random pruning (no SVF) | 70.68 ($\Delta$-3.56) | 70.98 ($\Delta$-4.32) | 81.59 ($\Delta$-3.71) |
> | No remasking | 73.11 ($\Delta$-1.13) | 73.41 ($\Delta$-1.89) | 84.09 ($\Delta$-1.21) |
> | Neither | 69.09 ( $\Delta$ -5.15) | 69.39 ($\Delta$-5.91) | 82.12 ($\Delta$-3.18) |
>
> SVF-guided pruning is the primary contributor (3.5–4.3pp over random pruning), remasking adds a consistent but smaller gain (1.1–1.9pp), and removing both causes the largest drop (3.2–5.9pp), confirming that the two mechanisms are complementary.
>
> **W2 & Q2: SVF analysis.**
>
> We note that SVF is only applied during the pruning window on intermediate completions $\hat{z}_0$, never on final completed sequences (final selection uses majority voting). Since these intermediate states have no final answer, we use Qwen3-235B-A22B-Instruct as proxy evaluator: its keep/prune decisions serve as reference labels, and we compute Precision, Recall, F1, Overlap@K, and Expected Calibration Error against SVF's decisions. We report the first/last SVF eval within the pruning window:
>
> | |Eval Point|F1 (K=2/4/8)|O@K (K=2/4/8)|ECE (K=2/4/8)|
> |-----------|-----:|---:|---:|---:|
> | GSM8K| First SVF | .676 / .696 / .731 | .392 / .468 / .597 | .330 / .281 / .231 |
> | GSM8K | Last SVF | .649 / .631 / .639 | .376 / .427 / .599 | .286 / .170 / .091 |
> | MATH-500 | First SVF | .650 / .697 / .704 | .343 / .470 / .557 | .306 / .267 / .204 |
> | MATH-500 | Last SVF | .634 / .629 / .631 | .349 / .424 / .590 | .258 / .158 / .097 |
>
> Because the pruning stage retains a fixed K, the number of positives is fixed, which makes Precision, Recall, and F1 numerically identical. P/R/F1 of 0.63–0.73 across all settings is well above random, confirming SVF provides a meaningful pruning signal. Importantly, ECE consistently improves from first to last SVF evaluation across all K and both benchmarks (GSM8K: 0.330→0.286, 0.281→0.170, 0.231→0.091; MATH-500: 0.306→0.258, 0.267→0.158, 0.204→0.097), indicating that SVF confidence becomes increasingly well-calibrated as denoising progresses and $\hat{z}_0$ predictions become more accurate.
>
> Notably, SVF results on MATH-500 are close to GSM8K despite MATH500's substantially lower base accuracy (26.4% vs. 67.6%), suggesting that SVF remains informative in harder settings rather than collapsing under model overconfidence.
>
> **W3 & Q6: Hyperparameter sensitivity.**
>
> We agree that the performance landscape can be steep around certain configurations. That said, this sensitivity manifests when deviating from the defaults; the recommended configuration ($W{=}[0.1, 0.6]$, $d{=}1.8$, $i{=}3$, $S{=}4$) itself performs consistently well across all 4 benchmarks and 3 models without task-specific tuning (e.g., MBPP default achieves 38.20%, surpassing Best-of-16 at 35.20%). For practitioners, we recommend adopting these defaults directly. Understanding why certain tasks are more sensitive to window placement is an interesting future direction. Regarding S=2 in Table 1: this is a typo and should be S=4, consistent with the appendix. We apologize for the confusion.
>
> **W4 & Q3: Final selection mechanism.**
>
> We checked our implementation: both Best-of-N and LLaDA-S use majority voting for final selection on all benchmarks. We will revise the paper to make this explicit and remove ambiguous wording.
>
> **Q4: SVF as final selection criterion.**
>
> Best-of-N uses majority voting for final selection. We evaluate replacing it with Top-1 SVF:
>
> | K | Majority Voting | Top-1 SVF | $\Delta$ |
> |-----|------:|------:|------:|
> | 2 / 4 / 8 | 74.24 / 75.30 / 85.30 | 70.98 / 72.73 / 74.85 | -3.26 / -2.57 / -10.45 |
>
> Majority voting consistently outperforms Top-1 SVF, with a particularly large gap at $K=8$. This is expected: majority voting aggregates answer-level information across all $K$ trajectories, while Top-1 SVF selects a single trajectory based on its reasoning coherence score. As discussed in W2, SVF is designed for intermediate pruning (eliminating weaker trajectories from a large pool) rather than final selection (picking the single best from a small, already-filtered pool).
>
> **Q5: Argmax vs sampling for SVF.**
>
> We use argmax. We compare the two strategies on GSM8K (LLaDA-8B):
>
> | $K$ | Argmax | Sampling | $\Delta$ |
> |-----|------:|------:|------:|
> | 2 / 4 / 8 | 74.24 / 75.30 / 85.30 | 73.18 / 74.17 / 84.70 | -1.06 / -1.13 / -0.60 |
>
> Argmax performs consistently better. We apologize for the confusion caused by Lines 216–219; we will make the description consistent with the implementation in the revision.
>
> **W5: Typo.**
>
> We will fix the Figure 8 caption to "LLaDA-S strategy trade-off between HumanEval Pass@1 and inference compute (NFE)" in the revision.

---

> > ### Author Rebuttal · Reviewer_UBqU · 2026-04-04
> >
> > Thank you for the thorough rebuttal. The component ablation (Q1) cleanly isolates SVF as the primary contributor and confirms complementarity with remasking. The SVF calibration analysis (Q2) with ECE metrics addresses the reliability concern; comparable F1 on MATH-500 and GSM8K despite the large base accuracy gap is convincing. The clarifications on selection mechanism (Q3), SVF as final selector (Q4), and argmax vs. sampling (Q5) resolve all remaining ambiguities. The committed text revisions (S=2 typo, Figure 8 caption, explicit selection mechanism description) should be carefully incorporated in the final version. Accordingly, I will raise my score.

---

> > > ### Author Response · Authors · 2026-04-06
> > >
> > > We are glad that our rebuttal has adequately addressed the reviewer's concerns. We sincerely thank the reviewer for raising the score and for the constructive engagement throughout the discussion. The discussion and feedback has been very helpful and has greatly strengthened the paper.

---

### Decision · Program_Chairs · 2026-04-30

**Decision:**

Accept (regular)

**Comment:**

This paper proposes LLaDA-S, a test-time scaling framework designed specifically for discrete diffusion language models. Starting from the observation that existing autoregressive-based test-time scaling algorithms are fundamentally incompatible with the parallel decoding nature of dLLMs, the authors integrate three mechanisms: (1) Hierarchical Trajectory Search, which dynamically redistributes computation during the early-to-mid denoising stages; (2) Self-Verified Feedback where the model evaluates its own intermediate completions without an external verifier; and (3) Partial remasking-based local branching, which explores diverse paths while retaining high-probability tokens. Across three dLLMs and four benchmarks, the framework achieves performance comparable to Best-of-N using 1.7–6.5× fewer Number of Function Evaluations.

The paper provides a practical solution to the timely and underexplored problem of test-time scaling for dLLMs, supported by thorough experiments across multiple models and benchmarks. Following the rebuttal, three out of four reviewers upgraded their scores to Accept with a "fully resolved" status, while the remaining reviewer maintained a Weak Accept, reflecting a positive consensus. Accordingly, the AC has decided to Accept the paper.

However, the authors must ensure the following commitments made during the rebuttal are reflected in the camera-ready version: (1) adding a discussion on the relationship with PG-DLM and SMC-based methods; (2) moving the SVF calibration analysis to the main text; and (3) correcting all typographical errors.